# Cytosolic actin isoforms form networks with different rheological properties that indicate specific biological function

Peter Nietmann[1,4], Kevin Kaub[1,2,4], Andrejus Suchenko [3], Susanne Stenz[1], Claas Warnecke[1], Mohan K. Balasubramanian [3] & Andreas Janshoff [1,2] ✉

The implications of the existence of different actins expressed in epithelial cells for network mechanics and dynamics is investigated by microrheology and confocal imaging. $\gamma$-actin predominately found in the apical cortex forms stiffer networks compared to $\beta$-actin, which is preferentially organized in stress fibers. We attribute this to selective interactions with $Mg^{2+}$-ions interconnecting the filaments' N-termini. Bundling propensity of the isoforms is different in the presence of $Mg^{2+}$-ions, while crosslinkers such as $\alpha$-actinin, fascin, and heavy meromyosin alter the mechanical response independent of the isoform. In the presence of myosin, $\beta$-actin networks show a large number of small contraction foci, while $\gamma$-actin displays larger but fewer foci indicative of a stronger interaction with myosin motors. We infer that subtle changes in the amino acid sequence of actin isoforms lead to alterations of the mechanical properties on the network level with potential implications for specific biological functions.

Actin constitutes a family of globular structural filament-forming proteins found in all eukaryotic cells. It is a major component of the cytoskeleton and one of the five most abundant proteins in eukaryotes[1]. The filaments are semiflexible, with an exceptionally large persistence length of $\approx$10 μm and with a diameter of only 7 nm enabling them to participate in ATP-driven contractile processes inside the cell, which are necessary to realize motility on various length scales[1–3]. Actin together with myosin and its many actin-binding proteins is pivotal for many important cellular processes, including muscle contraction, cell motility, adhesion, cell shape, cell division, and embryogenesis[4,5]. A common association with actin is its high degree of sequence conservation[6]. Sequence conservation is not only found in genes across different taxa but also extends to its different isoforms (Fig. 1). This is unexpected since such obvious redundancies should have disappeared during evolution, hence the existence of these almost identical isoforms should therefore be justified by selection pressure.

Two of the most closely related actin isoforms are cytosolic, non-muscle $\beta$-actin and $\gamma$-actin, which are encoded by different genes but produce nearly identical proteins except for four residues in their N-termini[6]. Despite their close similarity and their occurrence in the same cell types, it is proposed that the two isoforms are not redundant. For instance, it has been shown that $\beta$-actin and $\gamma$-actin localize to different parts of the cell and tend to incorporate into different actin cytoskeletal structures such as the cortex and stress fibers, respectively[6,7]. Furthermore, both isoforms appear to have dedicated roles in cell motility[7], epithelial-to-mesenchymal transitions[8–10], cellular junctions[11], and in hearing[12,13].

While this is evidence for the importance of isoform-specific amino acid sequence (AAS) for differences in cellular function, we do not understand how these nearly identical proteins perform such different roles. Previously, it was shown that in contrast to $\gamma$-actin, only $\beta$-actin mRNA gets spatially targeted to the cell periphery[14]. However, the reason for this behavior is probably not to be found in the AAS, but rather in the genes, which differ significantly[15]. Recently, a gene-editing technique was used to modify the mouse $\beta$-actin gene such that it coded for the $\gamma$-actin protein instead. This led to vital mice with

[1]Institute of Physical Chemistry, University of Goettingen, Tammannstr. 6, Göttingen 37077, Germany. [2]Max Planck School Matter to Life, Max Planck Institute for Medical Research, Jahnstr. 29, Heidelberg 69120, Germany. [3]Warwick Medical School, University of Warwick, Coventry CV4 7AL, UK. [4]These authors contributed equally: Peter Nietmann, Kevin Kaub. ✉e-mail: ajansho@gwdg.de

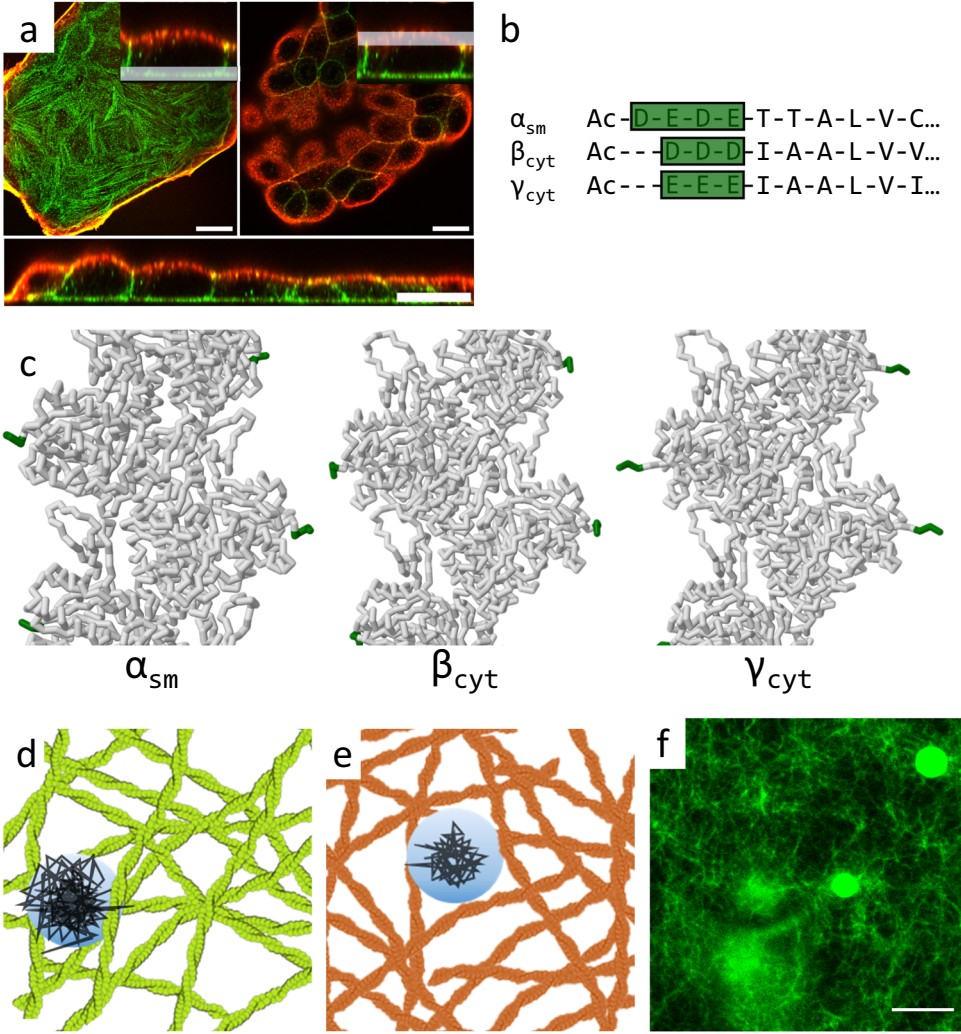

**Fig. 1 | β-actin (green) and γ-actin (red) are locally segregated in MDCK II cells.** β-actin is exclusively associated with basal stress fibers (**a**, left panel), while γ-actin is predominantly localized at the apical cortex (**a**, right panel). An orthogonal view (**a**, bottom panel) shows that the cell-cell boundaries mainly consist of β-actin[7] (scale bars: 20 μm). **b**, **c** The main differences (green) between the isoforms are concentrated at the N-terminus, which is located at very exposed positions along the filament (adapted from Cong et al.[109]). **c** Structures of actin isoforms measured via cryo-electron microscopy. For the shown protein structures the Protein Database (PDB) entries 3B5U, 8DNH and 8DNF were used[31,109]. **d**, **e** The schematics depict entangled isoactin filament networks (the green structures are β-actin filaments, red structures are γ-actin) with a microparticle (light blue) and different trajectories (black) indicating the measurement of the isoform-specific viscoelastic network properties. **f** Fluorescence micrograph of an γ-actin network (12 μM at 30 mM $Mg^{2+}$ stained with Alexa-Fluor-488-phalloidin) together with embedded microparticles (scale bar: 10 μm). Sample statistics are provided in Supplementary Tables 5, 6, and 20. The data underlying the figure is provided as Source Data.

unaffected phenotype suggesting that the two isoforms might be indeed interchangeable at least when exchanging β- with γ-actin[14]. In their seminal work, Allen and colleagues[16] investigated the mechanical characteristics of actin networks by employing a combination of isoforms. Their study revealed that the behavior of skeletal muscle α-actin and smooth muscle γ-actin tends to exhibit more elastic properties, in contrast to cytoplasmic β-actin. Nevertheless, because their experiments utilized blends of isoforms, the specific roles played by individual actin isoforms within the cell remain uncertain.

Here, we address the conundrum of whether different actin isoforms have indeed unique mechanical properties or are rather interchangeable. For this purpose, we examined equilibrium mechanics and ATP-driven structure formation of actin filaments on different length scales by forming reconstituted networks composed of pure isoforms.

We found that single filament behavior is largely conserved, while substantial differences in resilience and contractility between the different isoforms emerge when forming entangled networks. We observe that β-actin forms softer, less resilient networks with more locally dispersed contraction foci. By contrast, γ-actin builds stiffer

networks and the asters formed by myosin activity are larger, indicative of higher contractility compared to networks composed of β-actin. This goes hand in hand with higher connectivity as documented by larger stiffness of networks consisting of γ-actin and presumably stronger interactions of γ-actin with myosin motors. The study clearly shows that both actin-actin and actin-myosin interactions are altered depending on the exact AAS. We identified the crosslinking activity of $Mg^{2+}$ at physiological concentrations (around 2 mM[17–19] for free intracellular $Mg^{2+}$ ions and 10–20 mM[20–25] for the total magnesium concentration) as the main reason for the observed isoform-specific mechanics. Conversely, $Ca^{2+}$-concentration in the cytosol of eukaryotic cells is usually maintained at a concentration of 100 nM with the help of $Ca^{2+}$-ATP pumps. $Ca^{2+}$ influx increases the cytosolic concentration only to 10 μM, still very low compared to the free $Mg^{2+}$ concentration[26,27]. Bundling is suppressed in γ-actin networks, while β-actin shows a high bundling propensity. The interaction with divalent cations is also shown to be isoform-specific and partially resembles crosslinking. The potential biological consequences and implications are manifold. For example, the subtle balance between contractility

and percolation/crosslinking ability is important for rapid shape change by contractile actin cortices that allow cells to migrate. Moreover, the isoforms' structural reinforcement by $Mg^{2+}$-ions may also elucidate the favored localization of $\beta$-actin within stress fibers, owing to their tendency to form bundles, and $\gamma$-actin predominantly found in the cortex.

## Results

Most of the actin isoforms differ only by very subtle sequence variations. In particular, the differences between cytoplasmic $\beta$- and $\gamma$-actin are restricted to merely four amino acids located at the N-terminus, where the outermost amino acids are a sequence of three successive aspartate residues for $\beta$-actin and three successive glutamate residues for $\gamma$-actin, respectively.

Our aim was to explore whether these subtle sequence variations manifest noticeable structural, mechanical, and functional consequences for the behavior of F-actin within the network. These discoveries might eventually offer insights into the observed localization patterns of the different isoforms within the cell[28].

### Localization of actin isoforms in living cells

Figure 1a shows a confocal image after immunostaining of the two prominent cytoplasmic actin isoforms $\beta$- and $\gamma$-actin in a cluster of MDCK II cells (images of MDCK II monolayers and a SK-OV-3 cell in Supplementary Fig. 1). MDCK II cells have been used extensively to study the mechanical properties of the cytoskeleton, particularly the contribution of the cortex regarding resistance to external deformation[29,30]. The images confirm the distinct localization patterns of actin as previously predicted[7]. $\beta$-actin (green) is found exclusively in basolaterally forming stress fibers, while $\gamma$-actin (red) constitutes the apical cortex. This is in line with previous findings, which reported a similar spatial segregation pattern for MDCK II cells[7]. Figure 1b shows the AAS at the N-terminus for the three isoforms investigated in this study. Interestingly, the minor alterations in the N-terminal AAS determine how much the N-termini are exposed (Fig. 1c)[31]. In Fig. 1d, e, f, we show an illustration of the video particle experiments for different isoactins alongside a fluorescence image displaying a typical reconstituted actin network with enclosed beads.

### Isoform-specific network architecture and mechanics

Since actin isoforms occupy different locations within cells, the question emerged whether networks formed from pure isoforms already exhibit noticeable differences in the architecture of reconstituted actin networks. We also included $\alpha$-actin found in muscle cells as well as the so-called non-muscle (NM) actin, a mixture of $\beta$- and $\gamma$-actin, in this study since these are the most commonly used actin species in the literature. Interestingly, initial experiments conducted with an NM-actin network featured very similar properties compared to a network with the same constitution of $\beta$- and $\gamma$-actin (see Supplementary Fig. 3). Since NM-actin is a combination of two isoactins, the primary focus of this study centers on the individual $\alpha$-, $\beta$-, and $\gamma$-actin variants.

Figure 2a (left) shows confocal images of the three different F-actin networks in vitro. We did not observe any visually discernible distinctions among the actin networks formed by the different isoforms. This is not unexpected since entangled networks of semiflexible polymers show a universal scaling behavior in which the mesh size $\xi$ (calculated approximately to 390 nm for 12 $\mu$M actin[32], Supplementary Methods) depends on the concentration $c_A$ like $\xi \propto 1/\sqrt{c_A}$[33]. Despite their obvious structural similarity, we performed passive micromotion measurements of the different networks to investigate their mechanical properties (Fig. 1d, e, Fig. 2 center, b, c, d). For this purpose, we used one particle (1P) microrheology since the length scale we are interested in is on the order of the entanglement length $l_e$ and smaller

than the probe diameter $d$ ($l_e \approx \xi \approx 390$ nm, $d = 2.0 \pm 0.2\,\mu$m). The impact of bead diameter on the plateau modulus is shown in Supplementary Fig. 5. This length scale is responsible for the elastic properties of the network and, therefore well captured by 1P microrheology[34]. The viscoelastic spectra are obtained from Laplace transformation of the mean square displacement following an established standard procedure[35]. Typically, the plateau modulus that arises due to filament entanglement at intermediate time scales and comparable concentrations is in the range of 0.1 to 0.01 Pa. This is in good agreement with previous studies using solutions made from $\alpha$-actin[36]. A theoretical calculation of $G_0$ for 12 $\mu$M actin[37] results in a value of $\approx$22 mPa, which is in good agreement with our results (between $\beta$- and $\gamma$-actin). Surprisingly, the comparison of the mean spectra clearly shows that $\beta$-actin forms substantially softer networks with a plateau modulus ($G_0 = (7 \pm 1)$ mPa) almost one order of magnitude smaller than that of networks formed by $\gamma$-actin ($G_0 = (39 \pm 3)$ mPa). These differences are not limited to a specific concentration of actin. The power law behavior of the plateau modulus follows an expected $G_0 \propto c^{7/5}$ for each isoform well (Fig. 2d)[37–39], but they are shifted between the different isoactins as expected from the stiffness analysis (Fig. 2b). Broadly speaking, the observed power-law behavior affirms the presence of semiflexible filaments that constitute an entangled network, with contour lengths and a persistence length surpassing the mesh size, as indicated in previous studies[40,41].

We also conducted two-particle microrheology (2P-MR, Supplementary Fig. 2) to verify that our plateau moduli obtained from the viscoelastic spectra at 0.1 Hz are largely reporting on the bulk properties of the actin network rather than its local properties close to the bead (Supplementary Table 1). Our findings align with the results previously reported by Gardel et al.[34], who observed that the plateau moduli $G_0$ extracted from low frequencies are nearly indistinguishable, while on shorter time scales the methods substantially deviate. Along these lines, 2P-MR displays enhanced viscoelastic relaxation at intermediate frequencies consistent with diffusion over the filament length $L$[42]. Hence, 2P-MR can be rather compared with bulk rheology, with the latter requiring a significantly larger amount of material (see Supplementary Fig. 6 and Supplementary Table 2 for a comparison of different techniques). In fact, the 2P-MSD scales differently than the 1P-MSD, with MSD $\propto \tau^{1/2}$, while the 1P-MSD varies only little with $\tau$ and does not show a pronounced $L$-dependency of the response. However, it should be acknowledged that the data derived from 2P-MR exhibits poorer statistical reliability when compared to that of 1P-MR, particularly at the longer time scales we are interested in. As a consequence, we predominantly used 1P-MR in our study.

It has been shown that the interaction of myosin with actin depends on the particular isoforms[31,43,44]. The interaction is usually characterized by ATPase activity or duty cycle, i.e., the fraction of the ATPase cycle in which the motor protein remains tightly bound to an actin filament. This strongly influences contraction and thereby generation of tension. For example, myosins found in muscles have a low duty ratio (i.e., the myosins remain bound to the actin for a shorter amount of time) allowing for rapid muscle contraction, while myosins present in the cytoplasm feature a higher duty ratio[43]. As such, the cytoplasmic myosins facilitate a build-up of tension. With this in mind, the distinct actomyosin activities will probably translate into actin isoform (and also myosin isoform) specific contraction dynamics establishing different emerging network structures[44].

Addition of myosin to purely entangled actin networks resulted in the appearance of contraction foci, referred to as asters, at low ATP concentrations[45]. We distinguish the aster formation between the isoforms by aster size and aster connectivity, which show significant variations depending on the isoform (Fig. 3). Both $\alpha$- and NM-actin are characterized by a sparse number of asters. The aster size in the case of $\alpha$-actin is generally much larger than for the cytoplasmic actins suggesting differences in the duty ratios or larger connectivity,

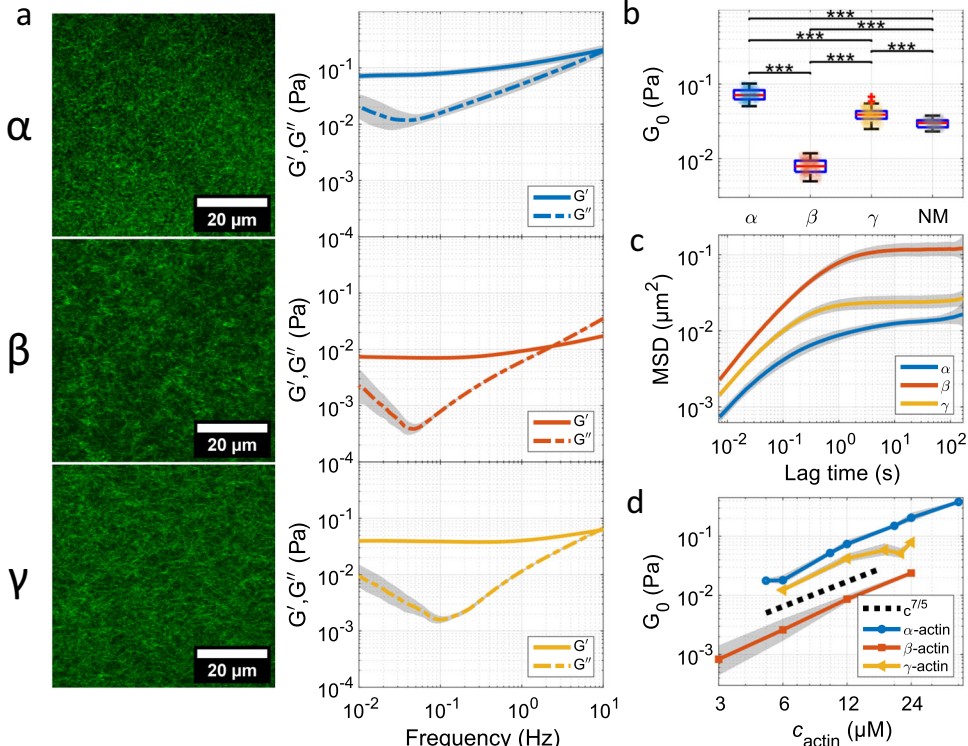

**Fig. 2 | Architecture and mechanical properties of networks formed by various actin isoforms (α-actin, β-actin, γ-actin).** (**a**, left) Confocal images of entangled actin networks (12 μM) formed by the three pure isoforms. (**a**, right) Median viscoelastic spectra of the actin networks obtained from passive microrheology measurements with SD. **b** Plateau moduli $G_0$ of the various isoforms and NM-actin obtained from the purely elastic frequency regime of the viscoelastic spectra at a fixed frequency of 0.1 Hz (compare Supplementary Table 1). Each transparent dot represents the median of a measurement. In each boxplot, the central thick red line represents the median, the blue box represents the first and third quartiles (the 25th and 75th percentiles), and the whiskers extend no further than 1.5 times of the distance between the first and third quartiles. $p$-values were determined by the Wilcoxon rank sum test. *$p \leq 0.05$, **$p \leq 0.01$, ***$p \leq 0.001$. **c** Averaged mean squared displacements of the three isoforms show strong caging and subdiffusion (**d**) Confirmation of the expected power law behavior of the plateau modulus $G_0 \propto c^{7/5}$ for each isoform by measuring $G_0$ for different actin concentrations. Measurement and sample statistics are provided in Supplementary Tables 7 and 8. CLSM micrographs and viscoelastic spectra for NM-actin can be found in Supplementary Fig. 4. The data underlying the figure is provided as Source Data.

respectively (Fig. 3a, b)[46]. By contrast, β-actin displays the smallest contraction foci, shortest distance, and highest density (or area fraction, Fig. 3 b, c, d).

The finding that actin networks composed of β-actin filaments display smaller contraction foci and a higher density in the presence of myosin can be explained by considering the phase diagram formed by contractility and connectivity as proposed by Bausch and coworkers and refined by Koenderink and coworkers. The authors suggest that higher connectivity of the network also leads to a higher propensity for global collapse at preserved motor activity[47–49]. If motor forces exceed the unbinding threshold of the crosslinks or entanglement points, these constraints eventually fail across the entire network. Well-connected actomyosin gels respond to motor activity at low ATP concentrations by undergoing a global contraction, where large asters emerge with very high actin density. However, depending on the connectivity, contraction events do not necessarily span system length scales. If motor activity remains constant, a local breakdown is expected for less interconnected networks, as found in our case for networks composed of β-actin. At some point, higher connectivity can also prevent contraction[50]. It is also conceivable that myosin binds more tightly to γ-actin rather than β-actin since the N-terminus of actin is a potential binding site for myosin[51].

Contraction foci are also frequently observed in vivo such as those found in the cortex of developing *C. elegans* embryos and the cell equator of mitotic cells[52]. It was postulated that contraction of the actomyosin cortex affects the spatial distribution of lipid-tethered proteins and thereby influences signaling[49]. The fact that γ-actin is predominately found at the cortex together with its ability to generate a

well-connected network with substantial contractility enables cells to rapidly change their shape, morphology, and stress response if required.

## Properties of single actin filaments

To explain the different, isoform-specific mechanical properties of purely entangled networks, a number of conceivable reasons for this behavior need to be considered. On the one hand, mechanical properties of entangled, semiflexible polymer solutions depend on the molecular dimensions of the filament, i.e., its diameter $d$, contour length $L$, the persistence length $l_P$ and on the other hand on the mesh size $\xi$ of the network[40,41,53,54].

First, we propose that the variations in the AAS could result in isoform-specific differences in the persistence length of actin filaments, which reflects the filaments' bending stiffness. Second, the variations in AAS and subsequent post-translational modifications may give rise to distinctive surface charge distributions, potentially leading to unique interactions and friction between filaments. The former effect would result in altered binding affinities to ABPs[55,56]. However, the emergence of attractive interactions among entangled chains would extend beyond the conventional treatment of entangled semiflexible chains. This might necessitate the introduction of the concept of transient crosslink-like interactions between filaments, potentially leading to the emergence of a glassy behavior[57,58], a phenomenon commonly observed in living cells[59].

## Filament length distribution

The filament lengths of a semiflexible chain forming an entangled network are expected to have only a minor effect on the plateau

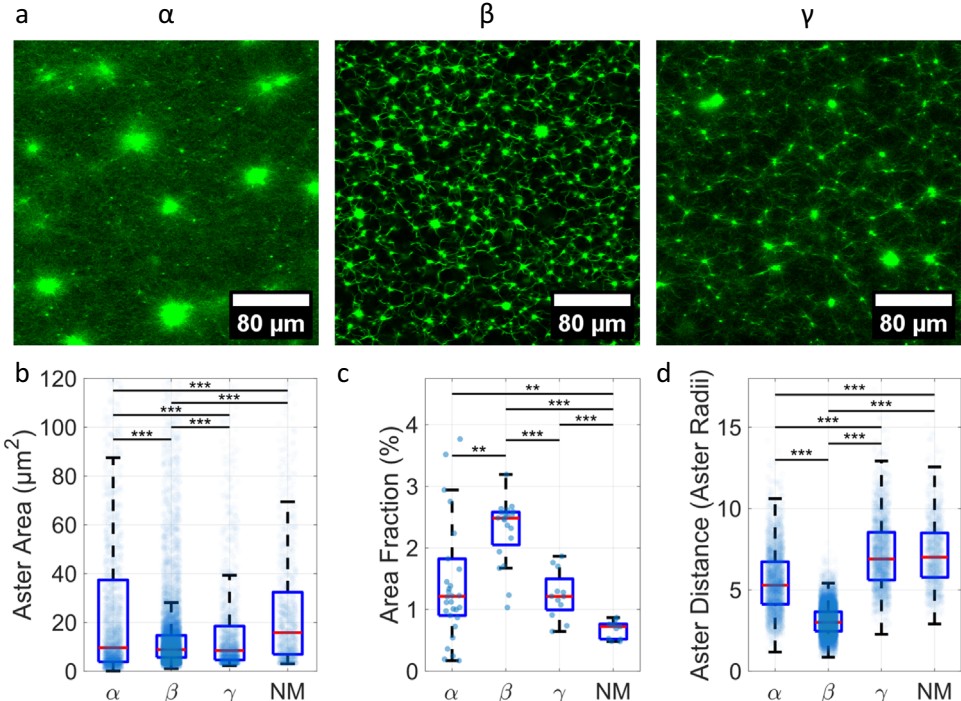

**Fig. 3 | Contractility of actomyosin networks composed of different isoforms.** Actomyosin networks constituted from different isoforms display significant differences in their appearance (**a**). Small clusters (asters) of actomyosin are visible in all samples, however, their size distribution (**b**) and area occupation (**c**) largely depend on the isoform. Asters deplete their surrounding of actomyosin, however, this depletion depends on the range of myosin-mediated pulling and on the size of the aster (**d**). Each transparent dot represents either a single aster (**b**, **d**) or an image (**c**). In each boxplot, the central thick red line represents the median, the blue box represents the first and third quartiles (the 25th and 75th percentiles), and the whiskers extend no further than 1.5 times of the distance between the first and third quartiles. $p$ values were determined by the Wilcoxon rank sum test. *$p \leq 0.05$, **$p \leq 0.01$, ***$p \leq 0.001$. Measurement and sample statistics are provided in Supplementary Tables 9–12. The data underlying the figure is provided as Source Data.

modulus at low frequencies where curvature stress dominates[39,60]. The general trend is that the longer the filaments, the stiffer the network[61,62]. At intermediate frequencies, bulk rheology experiments reveal a transition regime where the mechanical response is highly dependent on the filament length[42]. This regime is typically not captured in one-particle rheology experiments in which the bead size is smaller than the contour length.

We measured the filament length of surface-adhered filaments and found that the majority of actin monomers are incorporated in long filaments. For this we used a previously published method[63] to confirm a weighted filament length mean of 10 μm for $\beta$- and $\gamma$-actin and 5 μm for $\alpha$-actin, which is in good agreement to previous results[63] (Supplementary Fig. 8 and Table 1). The commercially available NM-actin mixture features shorter filaments on average (with a weighted mean of 3 μm). The distribution of lengths is roughly exponential with a variance that depends heavily on the specific isoform: while the pure isoforms $\beta$- and $\gamma$-actin display a rather large variance presumably due to inhomogeneous nucleation processes, the commercially available actins, especially NM-actin show rather small variations in contour length. This indicates different and isoform-specific nucleation dynamics. Furthermore, we conducted experiments with entangled actin networks supplemented with gelsolin to control the filament length and confirmed only a minor influence of different filament lengths on $G_0$. In fact, Liu et al.[42] showed that the 1P-MSD remains largely unaffected as long as the average filament length is substantially larger than the probe diameter. As such, we conclude that the filament length is not responsible for our observations (compare Supplementary Fig. 10).

**Persistence length**

The persistence length of semiflexible filaments is a material parameter that provides direct information about the mechanical properties of networks, and its changes could therefore be responsible for the observed isoform-specific network mechanics. Although most polymer models of semiflexible polymers predict a rather weak dependence of the plateau modulus on the persistence length $G_0 \propto l_P^{-\frac{1}{3}}$[60] it is known from actin-binding proteins and peptides such as phalloidin that alterations of $l_P$ can be substantial. Here, we employ imaging of single, fluorescently labeled filaments to determine their persistence length of the different isoforms. To visualize the filaments and also estimate the known[64] effect of labeling on their persistence length, we used different approaches. In order to determine the persistence lengths of the various isoforms and mixtures we filled phalloidin-labeled filaments into a thin (2 μm) chamber to confine fluctuations to 2D. This was necessary to avoid undulations perpendicular to the image plane that would otherwise lead to a smaller apparent contour length in 2D images. We inferred the persistence length from images as described elsewhere[2,65] and refer to this method as *angular cosine correlation*. All four phalloidin-labeled actins show very similar persistence lengths around 16 μm, which is in good agreement with previous results for $\alpha$- and NM-actin (Table 1)[2,64,66]. Therefore, we rule out that the contribution of persistence length to network stiffness explains the major difference in $G_0$ between $\gamma$- and $\beta$-actin.

**Filament-filament-interactions**

Since we could rule out the contour and persistence length of individual filaments as a possible reason for the substantial stiffness differences, we next considered sequence-dependent filament-filament interactions. The major difference in AAS between $\alpha$- and $\beta$-actin is found at the N-terminus of the protein, which is located at the protrusions of the outer filament[6,67,68] (Fig. 1). It also increases the charge density at this position, which may be relevant for network formation

**Table 1 | (a) Persistence length $l_P$ obtained from the angular cosine correlation of the different isoforms with a phalloidin label. Further information on persistence length determination is provided in Supplementary Fig. 7 and Supplementary Table 3. (b) Weighted mean $L_w$ and standard deviation of filament lengths in accordance to ref. 63. Compare Supplementary Table 4. Histograms of filament lengths are found in Supplementary Fig. 8. (c) Plateau moduli $G_O$ and standard deviations of commercially available $\alpha$- and NM-actin, as well as rhodamine-labeled versions of $\alpha$- and NM-actin using F-actin concentrations of 12 µM. The results for rhodamine-labeled actins are shown in Supplementary Fig. 9. Measurement and sample statistics are provided in Supplementary Tables 25–27. The data underlying the table are provided as Source Data**

| (a) | |
|---|---|
| Isoform | $l_P$ (µm) |
| $\alpha$ | $15.4 \pm 0.4$ |
| $\beta$ | $16.4 \pm 0.5$ |
| $\gamma$ | $16.8 \pm 0.4$ |
| NM | $15.5 \pm 0.4$ |

| (b) | |
|---|---|
| Isoform | $L_w$ (µm) |
| $\alpha$ | $5.2 \pm 3.0$ |
| $\beta$ | $10.3 \pm 6.7$ |
| $\gamma$ | $12.8 \pm 10.9$ |
| NM | $3.1 \pm 2.5$ |

| (c) | |
|---|---|
| Isoform | $G_O$ (mPa) |
| $\alpha$ | $83 \pm 13$ |
| NM | $33 \pm 4$ |
| Rhod-$\alpha$ | $748 \pm 115$ |
| Rhod-NM | $986 \pm 468$ |

and potential friction between filaments[28,56,69]. We propose that the associated attractive electrostatic interactions mediated via magnesium ions take the role of weak, transient crosslinks and therefore increase network stiffness just like a dedicated crosslinker protein such as fascin. This changes the properties of the network mechanics dominated by entropic stretching. Furthermore, the relevant length scale switches from the entanglement length to the distance of crosslinks. While the elastic modulus $G_O$ of entangled actin solutions in the absence of crosslinker molecules can be described by the partial suppression of filament fluctuations by surrounding polymers, this view fails to describe the mechanical response of the gel if the average distance between crosslinking points becomes smaller than the F-actin persistence length[70]. The interfilament interaction can also be explained in the framework of the so-called glassy wormlike chain (GWLC) rheology, in which an exponential stretching of the relaxation spectrum of the wormlike chain (WLC) model is added[71]. This stretching is achieved by only a single interaction parameter, which can be envisioned as a characteristic scale for the free energy barriers retarding the relaxation of the test chain's long wavelength eigenmodes.

To demonstrate how even subtle changes in charge density at the N-terminus of actin can alter network mechanics, we used rhodamine-labeled reconstituted $\alpha$- and NM-actin filaments that possess positively charged residues in the relevant pH regime associated with random lysine residues on the monomer surface[64,72]. These charges permit to invoke electrostatic interactions of the N-terminus similar to the magnesium-mediated attraction in non-labeled actin. We compared

their network stiffness with unlabeled entangled networks and found that the network stiffness, represented by the plateau modulus $G_O$, increased significantly after the introduction of rhodamine (Table 1). This indicates that small changes in the amino sequence, which lead to small differences in the surface charge structure of the filaments, alter the filament-filament interaction and eventually affect the network stiffness considerably.

### Electrostatic interactions between actin filaments

Since actin filament interactions in the absence of crosslinkers are presumably based on electrostatic interactions mediated by divalent cations like $Mg^{2+}$ (refs. 73–76), we investigated how a variation of the $Mg^{2+}$ or the $Ca^{2+}$ concentration affects the stiffness of isoactin networks. Filament bundling by counterion condensation is well known to occur at a threshold of 20–30 mM $Mg^{2+}$ for $\alpha$-actin[20]. We therefore also examined the threshold concentration, at which bundling begins for the other isoforms.

Figure 4 shows the results from bundling analysis of $\alpha$-, $\gamma$- and $\beta$-F-actin networks as a function of the $Mg^{2+}$ concentration. The bundling parameter $n_B$ denotes the number of filaments per bundle and shows a sudden increase at a specified $Mg^{2+}$ concentration, which is significantly different for the three isoactins. While filament networks of $\alpha$- and $\beta$-actin start collapsing and begin to form higher order bundle structures at 30 mM and 25 mM $Mg^{2+}$, respectively (Fig. 4a), $\gamma$-actin still polymerizes into a stable network of filaments until a concentration of 40 mM is reached (Fig. 4a). The network architecture is also quite different for the isoforms. $\beta$-actin bundles into thick cables with a high bundling parameter ($n_B \approx 150$) and large mesh sizes (inset Fig. 4b), while networks formed by $\gamma$-actin are more filigree and interconnected with smaller bundling propensity ($n_B \approx 80$) and smaller mesh sizes (Fig. 4b, compare upper and lower right image). The bundles formed by $\alpha$-actin display a similar mesh size compared to $\beta$-actin-bundles with a smaller bundling number ($n_B \approx 50$). The smaller bundle meshes of $\gamma$-actin networks are an indication of higher orthogonal connectivity. This is also shown for 40 mM $Mg^{2+}$ in $\gamma$-actin networks, but not for bundled $\beta$-actin networks (Fig. 4b and Supplementary Fig. 15), where the probe particles can pass almost unhindered through network meshes and continuum assumptions for passive microrheology are not met anymore. It is further confirmed by filament network stiffening with increasing $Mg^{2+}$ concentration. Plateau moduli continuously increase from 5 to 20 mM for $\alpha$- and $\gamma$-actin (Fig. 4c) with a steeper increase for $\gamma$-actin ($\approx 3\times$). For these isoactins $Mg^{2+}$ seems to have an effect comparable to regular, protein-based crosslinkers. By contrast, the mechanical properties of $\beta$-actin networks are not affected at the same $Mg^{2+}$ concentrations. Plateau moduli ranging from 5 to 15 mM (Fig. 4c) do not show a noticeable increase. At a threshold concentration (>20 mM), where bundling begins to occur, the networks appear to become softer due to depletion of actin around the micro probes (Fig. 4c). It is well known that other divalent cations also induce F-actin bundling[20,77]. Hence, we also investigated the influence of $Ca^{2+}$ on actin network mechanics and bundling (Fig. 4d). The plateau moduli increase with increasing $Ca^{2+}$ concentrations between 0.5 and 5 mM for $\gamma$-actin similarly to Fig. 4c, but not for $\alpha$- and $\beta$-actin, which are not affected significantly (compared to networks polymerized at 2 mM $Mg^{2+}$, which is shown at a $Ca^{2+}$ concentration of 0* mM). Bundling occurs at $\approx 10$ mM for all three isoforms (compare Supplementary Fig. 16). A remarkable difference compared to the $Mg^{2+}$ networks is found at a concentration of 0.5 mM $Ca^{2+}$, where all three isoforms are significantly softer than filament networks polymerized at 2 mM $Mg^{2+}$ and also at 1 mM $Ca^{2+}$ ($\approx 10\times$). This is an indication for incomplete polymerization of the network. Physiological concentrations are typically approximately 100 nM and can be as high as 10 µM if $Ca^{2+}$ influx occurs[78–82]. It is therefore reasonable to assume that magnesium plays a relevant role in actin network

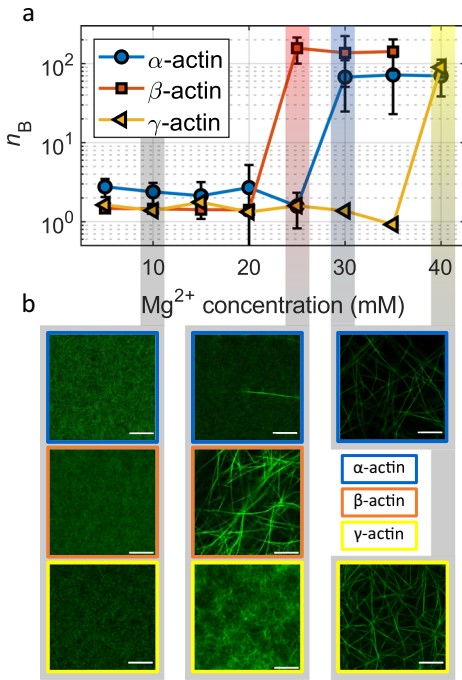

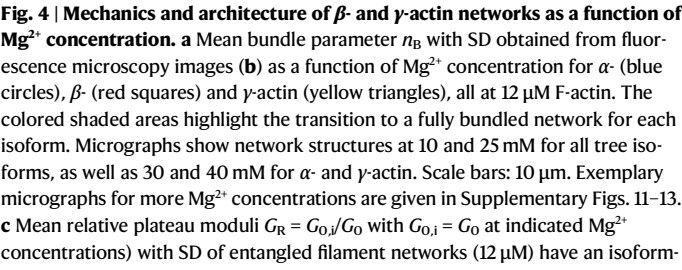

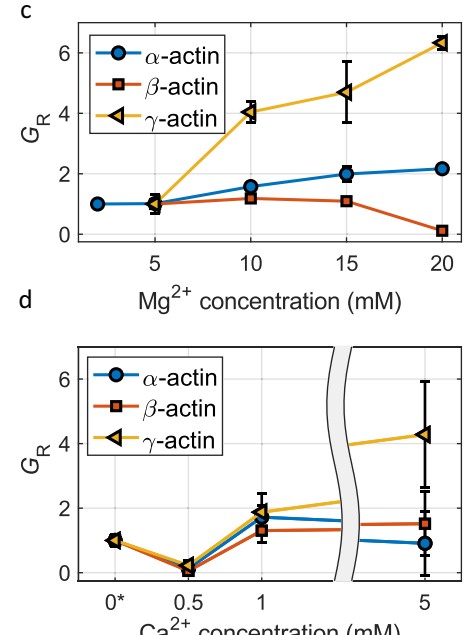

**Fig. 4 | Mechanics and architecture of β- and γ-actin networks as a function of Mg$^{2+}$ concentration. a** Mean bundle parameter $n_B$ with SD obtained from fluorescence microscopy images (**b**) as a function of Mg$^{2+}$ concentration for α- (blue circles), β- (red squares) and γ-actin (yellow triangles), all at 12 μM F-actin. The colored shaded areas highlight the transition to a fully bundled network for each isoform. Micrographs show network structures at 10 and 25 mM for all tree isoforms, as well as 30 and 40 mM for α- and γ-actin. Scale bars: 10 μm. Exemplary micrographs for more Mg$^{2+}$ concentrations are given in Supplementary Figs. 11–13. **c** Mean relative plateau moduli $G_R = G_{0,i}/G_0$ with $G_{0,i} = G_0$ at indicated Mg$^{2+}$ concentrations) with SD of entangled filament networks (12 μM) have an isoform-

specific slope at increasing Mg$^{2+}$ concentrations. α- and γ-actin show an increase (which is ≈3× steeper for γ-actin), while a decrease is found for β-actin. The moduli were normalized to the plateau modulus at 2 mM Mg$^{2+}$. **d** Mean averaged relative plateau moduli of entangled filament networks (12 μM) as a function of Ca$^{2+}$ concentration with SD. Plateau moduli are again normalized to the plateau modulus at 2 mM Mg$^{2+}$ represented by 0* (the concentration of Ca$^{2+}$ is 0 mM). Additional information for NM-actin can be found in Supplementary Fig. 14. Measurement and sample statistics are provided in Supplementary Tables 13–16. The data underlying the figure is provided as Source Data.

mechanics, while calcium does not act as a crosslinker at physiological concentrations.

In summary, our results show that β-actin is more prone to counterion condensation induced by magnesium ions than γ-actin. The higher bundling propensity of β-actin might explain the preferential location of β-actin in the basal plane of cells forming stress fibers. The fact that γ-actin remains as a dense filamentous network even in the presence of high Mg$^{2+}$ concentrations would be important for the structural integrity of the cortex. Similarly, the property of changing the network stiffness with a small variation of Mg$^{2+}$ as well as high Ca$^{2+}$ concentrations over orders of magnitude can also be a beneficial property for cortical structures. This would enable cells to generate prestress without global collapse of the network even at low crosslinker levels.

**Comparison to protein-based crosslinkers**

Finally, we address the question of how the differences between isoactins regarding their electrostatic interactions with cations might compare to conventional crosslinkers and whether we see further differences between β- and γ-actin. This is especially interesting for the evaluation of Mg$^{2+}$ ions' role in the mechanical properties of the iso actins and our interpretation of them as crosslinkers in the case of γ-actin.

We thus created crosslinked actin networks with a variety of crosslinkers, such as α-actinin, fascin and heavy meromyosin (HMM) in its rigor state, as well as 2, 7.5 and 15 mM Mg$^{2+}$ (Fig. 5 and Supplementary Figs. 17–19)[83]. For the chosen concentrations of fascin and α-actinin we do not observe a strong increase (much smaller than an order of magnitude, as previously reported[84,85]) in the plateau modulus

compared to the purely entangled networks of all isoactins. However, we do observe a noticeable impact upon addition of HMM, especially in the case of β- and γ-actin networks[86]. Similarly, we measured a strong increase in $G_0$ for γ-actin with the addition of Mg$^{2+}$, while β-actin and α-actin were not affected.

α-actinin and fascin are known to create parallel bundles of filaments[4,83]. Due to the relatively low ratio of crosslinkers to actin ($R = c_{crosslink}: c_{actin} = 1:100$), we do not expect a noticeable increase of the mesh size. Thus, the small impact of bundling crosslinkers like α-actinin and fascin on the network stiffness seems reasonable. Conversely, HMM presumably locates at the crossing points of intersecting filaments and interconnects them there[86]. This leads to more homogeneous, isotropically crosslinked networks. Our previous findings suggest an isoform-specific role for Mg$^{2+}$ cations, in which Mg$^{2+}$ acts as a bundling agent preferentially for β-actin, while γ forms a mesh-like network that stiffens with increasing Mg$^{2+}$-concentration. This effect is further pronounced in additional experiments conducted with the same crosslinker concentrations at elevated Mg$^{2+}$ concentrations (Supplementary Figs. 17–19). In the presence of 7.5 mM Mg$^{2+}$ together with either fascin or α-actinin β-actin forms thick bundles while γ-actin starts to form bundles at 15 mM Mg$^{2+}$. Concomitantly, we observe the initiation of bundles for α-actin at 7.5 mM for α-actinin, but not for fascin.

Therefore, we suggest that γ-actin forms transient crosslinks at the intersection of filaments by complexation of Mg$^{2+}$ between the N-termini. As such, the effect of Mg$^{2+}$ on γ-actin would resemble the architecture of HMM crosslinked networks. Conversely, for β-actin, the impact of Mg$^{2+}$ is similar to crosslinkers such as fascin and α-actinin forming bundles (Fig. 5).

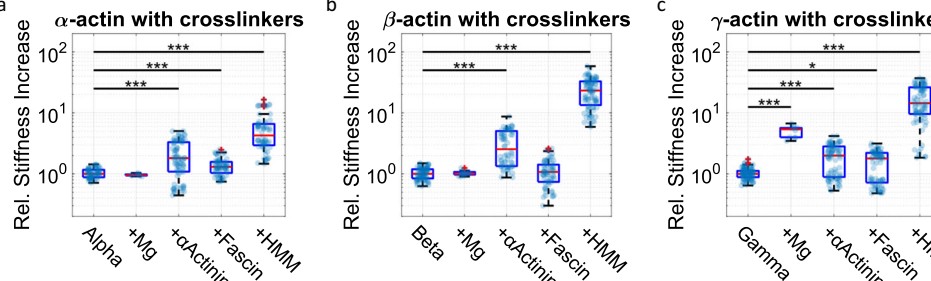

**Fig. 5 | Mechanics of different isoform-networks with various crosslinkers.** hVPT measurements reveal differences between crosslinked networks of $\alpha$-, $\beta$- and $\gamma$-actin (**a**, **b** and **c**, respectively; all networks at 12 µM actin). Since the purely entangled networks (Fig. 2) displayed different base plateau moduli, we chose to normalize the stiffness increase upon supplementation with crosslinkers using this base value. Crosslinkers were $Mg^{2+}$ (15 mM), $\alpha$-actinin (120 nM), fascin (120 nM) and rigor state HMM (120 nM). Each transparent dot represents the median of a measurement. In each boxplot, the central thick red line represents the median, the blue box represents the first and third quartiles (the 25th and 75th percentiles), and the whiskers extend no further than 1.5 times of the distance between the first and third quartiles. $p$-values were determined by the Wilcoxon rank sum test. $*p \leq 0.05$, $**p \leq 0.01$, $***p \leq 0.001$. Data for NM-actin are shown in Supplementary Fig. 4. Measurement and sample statistics are provided in Supplementary Tables 17, 18. The data underlying the figure is provided as Source Data.

## Discussion

Actins belong to a structurally conserved family of proteins found in all eukaryotic cells. They are mainly involved in the generation of contractile forces responsible for cell motility, cell shape, and mechanical stability. Because all mammalian actin isoforms share a high degree of sequence similarity, the existence of isoforms has been largely ignored by in vitro studies. As a result, $\alpha$-actin from skeletal muscle became the most studied actin isoform[87] because of its abundant availability and the assumption of interchangeability regarding its physical properties. However, even though muscle and cytoplasmic actin display relatively small sequential differences, it is well-conceivable that distinct structural and mechanical peculiarities exist[7,88]. One isoform constitutes large muscle tissues ($\alpha$-actin), while others ($\gamma$-, $\beta$-actin) form numerous types of filigree structures inside cellular confinements. Altogether, they fulfill a plenitude of roles ranging from structural integrity to cellular motion and signal transduction[7,8,10]. Additionally, there are many ABPs[4], among them also muscle and non-muscle myosin II, which specifically interact with only one of these isoforms[44]. Interchangeability was experimentally shown to some degree in cell experiments[12,15], but conversely also clear fatality of single knockouts[89]. A purely genetic and thus kinetic reason for different isoforms occupying overlapping spaces has also been proposed and shown to be conceivable[14,15]. Taken together, a clear, comprehensive picture is missing as it requires highly controlled conditions and a tailored bottom-up approach.

Here, we demonstrate that subtle changes of the sequence located at the N-terminus of $\alpha$-, $\beta$-, and $\gamma$-actin are indeed relevant for network mechanics. The N-terminus is part of an exposed arm of the 3D structure originating in the subdomain 1 of G-actin. This two-stranded helix is located on the filament exterior allowing the N-terminal region to be a site of interaction for myosin and several other actin-binding regulatory proteins[90]. The N-terminus is far removed from the monomer-monomer interaction sites and is known to play an important role in the recognition of actin-binding proteins, most notably myosin[31,91,92]. We quantified the properties of actin filaments formed from three isoforms across different length scales and states of organization, from single filaments to entangled or interconnected and eventually contractile networks. While the architectures of filament networks were indistinguishable, we found tremendous differences in equilibrium mechanics and network restructuring from non-equilibrium processes. Particularly, networks formed from $\beta$-actin turned out to be considerably softer than those formed from other isoforms. Considerable reasons for this behavior are variations of the persistence length, the filament length distribution, and the isoform-specific filament-filament-interactions (see Table 1).

Single filament properties could readily be excluded because we found no significant differences between the persistence and contour lengths of the different isoforms that would explain the differences in network stiffness. This was not unexpected in the first place since entangled networks show only a weak dependence of elasticity on the persistence length according to Isambert and Maggs[37,60]. However, it is well known that certain peptides and proteins might affect the bending stiffness of actin filaments quite substantially[64]. The filament length distributions of the different isoforms were also very similar and as a result, they were ruled out as a potential explanation for the distinct network properties observed.

Since we were able to largely eliminate individual filament properties as an explanation for the disparities in elasticity among the isoforms, we conducted a more detailed investigation into interfilament interactions. In this context, the leading contender for facilitating transient connections between filaments was the electrostatic interaction mediated by magnesium ions, akin to protein-based transient crosslinkers[93]. Increasing the concentration of $Mg^{2+}$ during actin polymerization and measuring the network stiffness as well as architecture revealed significant differences between $\gamma$- and $\beta$-actin. The network stiffness of $\beta$-actin measured as the plateau modulus $G_0$ agrees well with a theoretically calculated value purely based on hard-body interactions in a wormlike chain model[37]. It is also unaffected by the presence of $Mg^{2+}$ ions until filaments abruptly begin to align and form thick cable-like bundles above a $Mg^{2+}$ concentration of 20 mM. Very similar behavior has been reported for $\alpha$-actin[20]. Our measurements reveal that the bundling threshold is only slightly higher at 30 mM. In contrast, both $\alpha$- and $\gamma$-actin networks exhibit significant stiffening as the $Mg^{2+}$ concentration increases. Remarkably, $\gamma$-actin retains its filamentous structure even at a 30 mM $Mg^{2+}$ concentration, whereas $\beta$-actin is already bundled at this level of $Mg^{2+}$. Ultimately, when exposed to 40 mM $Mg^{2+}$, the $\gamma$-actin networks also initiate bundling. In contrast to $\beta$-actin, these bundles exhibit a more organized, interlinked network structure rather than forming thick, cable-like bundles.

Generally, polyvalent counterions can mediate a lateral attraction between two like-charged polyelectrolytes such as DNA or F-actin leading to bundling[73]. It is well known that weak interactions between F-actin and $Mg^{2+}$ ions originate from nonspecific trapping in the electric field surrounding F-actin due to its polyelectrolyte nature[94]. It was found that magnesium ions bind loosely and non-specifically to actin filaments, and thus show a behavior typical for counterions in polyelectrolyte solutions[94]. F-actin bundling in the presence of polyvalent cations emerges from competition between the attractive forces and the associated loss in entropy[77,95]. It was found by computer simulations that the attractive force results not only from the binding pattern

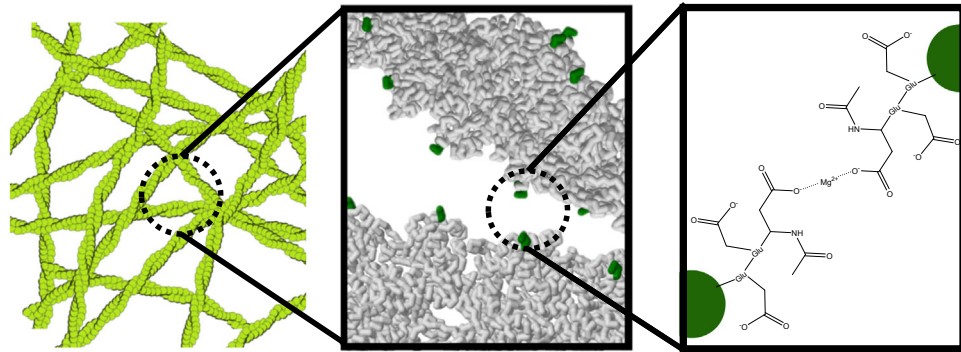

**Fig. 6 | Scheme illustrating transiently crosslinking of actin filaments (green structures in the left panel) by Mg$^{2+}$-ions.** Mg$^{2+}$-ions are anticipated to be chelated by the carboxylated residues at the N-terminus (middle and right panel).

correlations but also from the filament configuration correlations with fixed binding patterns. It turned out that the tightly bound counterion site is not affected by bundling in contrast to the medium and loosely bound ones[95]. This competition between binding and entropy also leads to the observation that above a critical contour length attractive interactions exceed entropy and bundling prevails[95]. The counterion-charge alone can not be the only determining variable controlling bundling of F-actin, since it is well known that bundling threshold concentrations differ between different divalent cations[20,77]. Our results on Mg$^{2+}$ in contrast to Ca$^{2+}$ also confirm this view (compare Fig. 4 and Supplementary Fig. 16). One would expect Mg$^{2+}$ to bind more tightly to carboxylated motifs due to its smaller size and higher charge density, however, certain Asp/Glu enriched motifs such as EF hands show substantially higher affinity for calcium than magnesium ions[96]. In their study, Jing et al. demonstrated that this phenomenon can be elucidated by considering the tight arrangement of numerous polarizable residues surrounding the metal ion.

The viscoelastic spectra and architecture of networks formed by $\gamma$-actin in the presence of an elevated Mg$^{2+}$ concentration suggest that magnesium ions act like transient crosslinkers (compare Fig. 6). The plateau modulus $G_0$ is substantially higher than expected for entangled filaments and the contractility of myosin also suggests a well-connected network. The difference in bundling propensity of $\beta$- and $\gamma$-actin requires us to also consider the specificity of the N-terminal forming complexes with a divalent cation. Both N-terminal amino acids Asp and Glu have been shown to form complexes with Mg$^{2+}$[97]. As such, the complexation of Mg$^{2+}$ cations between the singular N-termini of two adjacent actin filaments would serve as weak, transient crosslinks. Albeit the complex formation constants of magnesium ions with Asp or Glu are typically quite similar, it was found that the sidechains of Asp and Glu exhibit different pKa values for the exposed carboxyl groups in which aspartate is slightly more acidic[97]. The difference originates from the longer side chain of glutamate donating more electron density (inductive effect[98]) into the residual carboxyl group, which essentially destabilizes the negative charge. Conversely, the chain of Glu is longer and more exposed compared to Asp.

Hence, it can be anticipated that the binding of Mg$^{2+}$ cations by Glu will exhibit a preference over Asp. This conjecture finds support in the observed rise in $G_0$ at higher Mg$^{2+}$ levels in the case of $\gamma$-actin, whereas no such increase is observed for $\beta$-actin. If quasi-crosslinking, in the form of complexation, is indeed occurring, then an elevated concentration of crosslinkers should enhance interconnectivity, consequently leading to an increase in $G_0$.

A remaining question is why $\alpha$-actin generates a network stiffness similar to $\gamma$-actin but shows a bundling behavior more similar to $\beta$-actin. On the one hand, the first amino acid at the N-terminus of $\alpha$-actin is an aspartate and thereby should be more similar to $\beta$-actin (Fig. 1b). On the other hand, the charge density at the N-terminus of $\alpha$-actin is

substantially higher compared to $\beta$ and $\gamma$, since its fourth amino acid is also negatively charged in contrast to $\gamma$- and $\beta$-actin (compare Fig. 1b). At low magnesium concentrations (2 mM) the higher charge density of $\alpha$-actin N-termini localizes divalent cations more effectively and favors complexation of Mg$^{2+}$ compared to $\gamma$-actin (the plateau modulus of $\alpha$-actin is 2× higher than that of $\gamma$-actin, see Fig. 2b). At increased Mg$^{2+}$ concentrations (above 10 mM), this effect is compensated by the generally higher abundance of the cations, thus supporting more stable complexation $\gamma$-actin. Consequently, the overall network rigidity of $\gamma$-actin exceeds that of $\alpha$-actin. (compare Figs. 2b and 4c).

Calcium is the second most relevant divalent cation in a cellular context. We also investigated its influence on the mechanics and structure formation in reconstituted actin networks across various isoforms. Similarly to Mg$^{2+}$, it bundles F-actin in a process of counterion condensation at a certain threshold concentration of Ca$^{2+}$. In contrary to Mg$^{2+}$, this concentration is very similar for all actin isoforms (10 mM Ca$^{2+}$ at 12 µM actin). Significant network stiffening can only be observed for $\gamma$-actin (compare Fig. 4d). Ca$^{2+}$ concentrations below 1 mM (which is well above physiological concentrations[78–82]) lead to significantly softer, not yet fully polymerized actin networks. So the influence of Ca$^{2+}$ ions on the mechanical properties of F-actin networks at physiological concentrations (approximately 100 nM) can be assumed to be negligible, but the behavior at elevated concentrations still illuminates the nature of the interfilament interactions of actin. Given that there is no notable distinction among the isoforms in their bundling threshold, we can dismiss the notion that the highly isoform-specific initiation of Mg$^{2+}$ network bundling arises solely from charge-based factors. Instead, we can infer an isoform-specific mechanism.

The general importance of filament-filament interactions for the viscoelastic properties of the networks is further demonstrated by rhodamine-labeled $\alpha$- and NM-actin serving as a model system. The labeling exposes a positive surface charge by covalently binding rhodamine to a random lysine residue on the filament surface. This positive charge can be interpreted as a permanently bound cation reducing charge repulsion and increasing bundling propensity as well as enabling direct attractive electrostatic interaction with negatively charged N-termini. In our investigation, we observed a minimum ten-fold increase (as depicted in Supplementary Fig. 9) in the average plateau modulus when compared to networks without labeling. This finding indicates two important aspects: firstly, the significant involvement of ionic interactions in determining network stiffness, and secondly, it highlights the need for caution when conducting actin network experiments using rhodamine or similar labels in cases where network structure or mechanical properties are pertinent. The substantial impact on both factors could introduce considerable bias into the results.

We also examined how the quasi-crosslinking mediated by Mg$^{2+}$ compares to known actin crosslinkers, such as $\alpha$-actinin, fascin, and

HMM[4,83]. A concentration of 15 mM Mg$^{2+}$ did not alter the network stiffness for $\beta$-actin, but it noticeably increased the plateau modulus for $\gamma$-actin (Fig. 4). As such, for $\beta$-actin the impact of Mg$^{2+}$ is reminiscent of those of $\alpha$-actinin and fascin, leaning into the idea of a higher propensity of $\beta$-actin to form bundles, which increase the network mesh size. On the other hand, Mg$^{2+}$ affects $\gamma$-actin networks similarly to HMM, which indicates a tendency of $\gamma$-actin to remain in an isotropic network. From our results based on crosslinked networks, it can be concluded that both isoforms are fundamentally biased towards certain network architectures. This bias can also be phrased in terms of network connectivity, with $\beta$-actin displaying smaller connectivity to allow for easier bundling and $\gamma$-actin featuring higher connectivity to remain in an isotropically crosslinked network. To further challenge this interpretation, we also assessed the aster formation of ATP-driven actomyosin networks. We confirmed that higher connectivity is achieved with $\gamma$-actin by studying the contractility of the different isoforms. Smaller contraction foci (asters) are found for $\beta$-actin indicative of less connected networks, while the other isoforms display larger clusters that are expected for reasonably resilient networks.

We found further confirmation for quasi-crosslinking of $\gamma$-actin similar to HMM by increasing the Mg$^{2+}$ concentration in networks crosslinked by $\alpha$-actinin, fascin, and HMM. The general expectation is a reduction of the threshold concentration of Mg$^{2+}$ for bundling at least for fascin and $\alpha$-actinin due to cooperative bundling. Interestingly, in the presence of elevated concentrations of Mg$^{2+}$ networks of $\alpha$-actin and even more so $\gamma$-actin are stabilized in a filamentous structure against bundling compared to $\beta$-actin (see Supplementary Figs. 11–13, 17–19).

In summary, we present evidence for isoform-specific mechanical and structural properties of entangled actin filaments. Our findings reveal that $\beta$-actin exhibits a greater inclination to form bundles, whereas $\gamma$-actin tends to maintain a mesh-like structure. This architectural contrast in reconstituted networks aligns with the localization patterns observed in eukaryotic cells, where $\beta$-actin predominantly resides in stress fibers, and $\gamma$-actin is more commonly found in the cell cortex.

Given the significant level of conservation observed in actin, such a degree of precise adjustment does not come as a surprise. Eukaryotic life requires substantial adaptability to meet the various extrinsic and intrinsic requirements for proper function ranging from the response to environmental factors to the generation of motion, growth, and development. Actin is among the key molecules to enable such variability and responses and the presence of isoform with subtle sequence alterations is an important factor to provide the necessary variability. The minute differences between cytoplasmic actin isoforms regarding their translation dynamics and AASs seem to serve as the starting point for differential post-translational modifications, which in turn initiate the different emerging cytoskeletal substructures. Despite the complexity encountered in living cells, our results demonstrate the importance of subtle differences between isoforms and how the associated weak interactions on the molecular scale translate into substantial changes on larger length scales in network architecture, stiffness, and contractility.

## Methods

### Cell culture and immunostaining
MDCK II cells (Madin-Darby canine kidney; ECACC No. 00062107; authentication via ECACC; confirmed negative for Mycoplasma via PCR testing) and SK-OV-3 cells (ovarian adenocarcinoma; ATCC No. HTB-77; validated by ATCC; confirmed negative for Mycoplasma via PCR testing) were cultivated in minimal essential medium (MEM with Earl's salts, containing Glutamax™ (Gibco™, Thermo Fisher Scientific) and 10% fetal calf serum (FCS; BioWest, Nuaillé, France)). Stock solution and samples were kept at 37 °C and 5% CO$_2$ (Heracell 150i; Thermo Fisher Scientific, Waltham, MA).

Confluent cells were detached from culture flasks by addition of 0.25% trypsin and 0.02% EDTA (Biochrom). Cells were seeded into Petri dishes (Ibidi, Martinsried, Germany) and kept at 37 °C and 5% CO$_2$ for 24 h, unless stated otherwise. MDCK II cells were either seeded at a density of $1.5 \times 10^5$ cells per dish for obtaining confluent cell layers or less dense ($1.5 \times 10^4$) cells per dish for forming cell clusters.

Primary antibodies (anti-$\beta$-actin antibody, clone SP124; anti-$\gamma$-actin antibody, clone 2A3) were purchased from Sigma-Aldrich Chemie GmbH (Taufkirchen, Germany). Immunostaining was conducted in accordance to Dugina et al.[7]: prior to fixation, cells were rinsed with preheated (37 °C) MEM supplemented with 20 mM HEPES. Cells were then fixated in preheated (37 °C) MEM supplemented with 1% PFA for 30 min at 37 °C and 7.5 % CO$_2$. Afterwards, the cells were treated for 5 min with cooled MeOH (−20 °C) and were then washed 3× with PBS (20 °C). Fixed cells were then treated with 0.1% Triton X in PBS for 5 min, washed 3× with PBS, and then incubated for 30 min with blocking/dilution buffer (BDB) at 20 °C. The liquid was then aspirated from the cells and 200 μL BDB with primary antibodies for $\beta$-actin and $\gamma$-actin (dilution of 1: 200) was added on top of the cells and incubated at 20 °C for 1 h. Cells were then rinsed 3× with PBS and subsequently incubated with 200 μL BDB containing secondary antibodies purchased from Fisher Scientific GmbH (Schwerte, Germany) in a dilution of 1: 400 at 20 °C for 1 h.

### Sample preparation
All proteins (rabbit skeletal muscle: skeletal muscle $\alpha$-actin, rhodamine-labeled skeletal muscle $\alpha$-actin, myosin II; human platelet: non-muscle actin, a mixture of 15% $\gamma$- and 85% $\beta$-actin, also referred to as NM-actin, rhodamine-labeled NM-actin) except for pure $\gamma$- and $\beta$-actin (vide infra) were purchased from Cytoskeleton, Inc (Denver, CO).

**Skeletal muscle and commercial non-muscle actin.** G-actin (rabbit skeletal muscle and human platelet non-muscle) samples were prepared from lyophilized actin monomers by dissolving the powder in deionized water to obtain a concentration of 10 mg/ml. The samples were further diluted by the addition of G-buffer (2 mM Tris HCl, 0.2 mM ATP, 0.5 mM DTT, 0.1 mM CaCl$_2$, 0.01% NaN$_3$, pH 7.8). Carboxylated probe particles (2 μm in diameter, L4530, Sigma) were coated with BSA by incubation with BSA solution (1 mg/ml BSA in G-Buffer, Thermo Fisher Scientific) and subsequent washing with G-buffer (3×). The polymerization of G-actin to F-actin was achieved by adding a polymerization solution (20 mM MgCl$_2$, 1000 mM KCl, 50 mM EGTA, 0.01% NaN$_3$, pH 7.4), the added volume of polymerization solution was 1/10 of the final sample volume.

**$\beta$- and $\gamma$- cytoplasmatic actin.** Cytoplasmatic $\gamma$- and $\beta$-actin were obtained from the yeast *Pichia pastoris*[99]. A modified strain of yeast was created to express actin isoforms with specific post-translational modifications to approach a synthetic recreation of organism-specific proteins. The actin was stored at −80 °C in G buffer, and the initial concentration was determined with a NanoDrop™ 2000c (Thermo Fisher Scientific) UV/Vis spectrometer. Further dilution and polymerization were done the same way as for the commercial actin preparations. The degree of polymerization was verified by ultracentrifugation of polymerized networks and measurement of the residual concentration in the supernatant by UV/Vis spectroscopy (NanoDrop™).

### Microrheology
Passive microrheology data was collected via holographic video particle tracking (hVPT) using monochromatic light from a light-emitting diode ($\lambda = 660$ nm) in a bright-field setup with a 60× objective (CFI Achromat FF, numerical aperture [NA] = 0.80, working distance = 0.3 mm; Nikon) at 138 fps. Two-dimensional microparticle trajectories were recorded with commercial tracking software (AFS, Lumicks BV,

software: AFS-Tracking-G2-v1.1.5) in real-time. Particle traces were saved as technical data management streaming files (*.tdms). These were converted to MATLAB R2020b[100] files (*.mat) with a custom made code, accessing function originally published by Jim Hokanson[101]. MSDs were then calculated from the trajectories and Laplace transformed by accessing function originally published by the Kilfoil laboratory[102]. The result was plugged in the generalized Stokes-Einstein equation to obtain the complex shear modulus $G^{*}$.

To compare one-particle microrheology (1P-MR) with two-particle microrheology (2P-MR), we employed actin solutions with a concentration of 24 µM. To enhance the signal-to-noise ratio in the two-particle displacement correlation, our approach involved monitoring the motion of over 100 particles (each with a diameter of $a = 1\,\mu m$) using brightfield microscopy equipped with a 60× oil-immersion objective (Olympus IX 71, Evident, Japan). This involved capturing several thousand frames in a recording time of more than 1 min (at a frame rate of 43 fps). Subsequently, we averaged the data over multiple datasets (>10 sets each). With high precision, we identified the positions of particle centers in each frame, achieving sub-20 nm accuracy, using a Zyla sCMOS camera (Andor, Oxford Instruments, USA). Following this, we determined particle trajectories from the same sample using Trackmate in Image J. A self-written Python script (3.10) was used to compute the particle correlation as described in Crocker et al.[103].

### Mechanical sample properties−plateau modulus

The mechanical properties of the samples are characterized by their storage moduli $G'$ obtained at $\omega = 0.1\,s^{-1}$. These moduli are typically interpreted as the plateau moduli $G_0$, due to the close proximity of the most frequently observed minima in $G''$ and the shallow slope of $G'$ within this frequency range. The selection of this frequency can also be rationalized by considering the point at which the 1P-MSD and 2P-MSD converge, as illustrated in Supplementary Fig. 2 and pointed out by Gardel et al.[34]. The slight departure from commonly employed methods[104] can be justified by considering that the inter-particle variance in $G_0$ is greater than the errors stemming from this deviation, as demonstrated in Supplementary Table 1. Furthermore, this deviation aligns with the order of magnitude seen when comparing plateau moduli obtained from different measurement devices (Supplementary Table 2). Moreover, this approach facilitates a direct and model-free comparison of the data.

Shear moduli were calculated from mean squared displacements (MSDs), which were obtained from particle trajectories as described previously[105].

### Imaging

Images of filaments, networks, and cell structures were taken with a confocal laser scanning unit (CLSM, FV10, Olympus) connected to an inverted microscope (Olympus IX 83) equipped with a 100× oil-immersion objective (UPLFLN100XO2PH, Olympus).

### Measurement of persistence lengths and length distributions

F-actin with a concentration of 1 mg/ml was prepared and polymerized as described before[87] and then further diluted by a factor of 1000 with G-buffer to enable the observation of non-overlapping, single filament fluctuations.

Alternatively, the filaments were adhered and fixated to the surface of a coverslip enabling us to compare the bending stiffness of labeled and unlabeled filaments.

Microscope glass slides (#1, 60 × 24 mm², Carl Roth) were plasma-cleaned and coated with poly-D-lysine (PDL, Sigma-Aldrich). A diluted F-actin filament sample (1 µL) was carefully added to the slide. The actin sample was either pre-labeled (saturated with Alexa Fluor™ TM 488 phalloidin) or unlabeled. In the latter case, the phalloidin label was added after adhesion of the filaments. Finally, the droplet was covered with a saturated sucrose solution to reduce evaporation.

For filament length distributions images of filaments were taken with a confocal laser scanning unit (FV10, Olympus, software version: FV10-ASW 04.02.01.20) connected to an inverted microscope (Olympus IX 83) equipped with a 100× oil-immersion objective (UPLFLN100XO2PH, Olympus). Raw images were binarized and skeletonized with the ImageJ 1.44[106] plugin Ridge detection[107]. The resulting minimal filaments were combined into stacks and analyzed via the MATLAB-based software Persistence (De La Cruz Lab[65]).

### Measurement of bundle parameters

Bundle parameters were calculated from confocal images taken under standardized conditions (laser and imaging properties as well as distance from the glass cover slides and sample properties were kept comparable). Line profiles were baseline-corrected (the mean intensity between bundles was subtracted) and the mean intensity of a peak was divided by the intensity of a single filament to obtain the number of filaments per bundle, i.e., the bundle parameter $n_B$. A comparison with theoretically calculated average mesh sizes for entangled networks[32] for the same G-actin concentration was also considered to give an additional method to calculate $n_B$.

### Aster analysis

Reconstituted actin networks (12 µM) were polymerized together with myosin II (600 nM). The emerging actomyosin asters were imaged with a CLSM under standardized conditions. Images were further processed using the open-source software FIJI and a custom-written MATLAB program.

### Statistics and reproducibility

Statistical analysis was performed using a Wilcoxon rank sum test with $*p \le 0.05$, $**p \le 0.01$, $***p \le 0.001$. Data are presented as mean ± SD unless otherwise indicated. Information on reproducibility and number of independent measurements are provided in the Supplementary Tables 5−38.

### Reporting summary

Further information on research design is available in the Nature Portfolio Reporting Summary linked to this article.

## Data availability

Source data are provided with this paper. Additional data used in this study are deposited at gro.data.[108] Source data are provided with this paper.

## Code availability

Source code of self-written scripts are deposited at gro.data[108].

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

## Acknowledgements

This research was conducted within the Max Planck School Matter to Life supported by the German Federal Ministry of Education and Research (BMBF) in collaboration with the Max Planck Society (K.K. and A.J.). The Deutsche Forschungsgemeinschaft (JA963/19-1, EXC 2067-390729940) are gratefully acknowledged for financial support. We thank Burkhard Geil for fruitful discussions and providing Python code for two-particle microrheology. M. Balasubramanian acknowledges Wellcome Trust WT101885MA and Biotechnology and Biological Sciences Research Council (BB/S003789/1).

## Author contributions

P.N. and K.K. carried out the experiments, analyzed the data and wrote the manuscript. S.S. and C.W. performed confocal imaging and participated in microrheology experiments. A.S. and M.K.B. prepared and characterized the actin isoforms from yeast. They also reviewed and edited the manuscript. A.J. carried out the data analysis, planned the research, and wrote the manuscript.

## Funding

## Competing interests

The authors declare no competing interests.
