## [Peer Review File · Nature Communications]

REVIEWER COMMENTS

Reviewer #1 (Remarks to the Author):

This study addresses the important and still unresolved issue of why cells express different actin isoforms that differ only in a few acidic residues at their N- termini. The study employs a range of state-of-the-art methods to use microbiological methods to obtain shear moduli of networks formed by the different isomers, as well as imaging to determine their filament lengths, persistence lengths, and reorganization due to cross linkers, magnesium ions, and myosin motors. A difference is noted between gamma and beta actins, with the conclusion that these differences in network forming properties help explain why genes coding for these different actin isoforms are not interchangeable themselves. This protein-based interpretation is a contrast to the recent model suggesting that it is the RNA coding for these different isoforms, which are localized differently and translated at different rates, and not the protein structure itself that accounts for the retention of these actin isoforms during evolution.

Several technical issues, largely regarding the microbiology studies, need to be addressed before the significance of this study can be determined.

There is also an issue of novelty in the comparison of these different isoforms. The paper cites a review article by Ira Hermans but does not cite this lab's very closely related study, (Allen PG ... Herman IM. Phalloidin binding and rheological differences among actin isoforms. *Biochemistry*. 1996;35(45):14062-9. doi: 10.1021/bi961326g) which also measured the shear moduli of networks formed by beta and gamma actin, as well as a range of different alpha actins, along with the lengths and persistence lengths of the filaments used to form the networks. It is not clear why this work is not cited, as it covers much of the same ground and also finds that gamma actin is more efficient than beta actin at making elastic networks.

The major issue in understanding the significance of the microbiological data is the relationship between the mesh size and the average filament length. At the high dilution needed for microbiology, the mesh size is stated to be on the order of a micron, or the size of the probe (2 μm), but the average filament length for all isoforms, and especially for the non muscle actin, is very close to the same distance or even smaller. It is hard to see how a filament system with average length 0.85 microns can make an elastic network with the mesh size of 1 micron. Even if the mesh size is small enough for filaments of this length to overlap each other, the model of a network of semi flexible polymers, and especially as stated here entangled polymers, cannot apply. At a minimum, conventional measurements of elastic modulus at protein concentrations large enough so that the mesh size is much larger than the average filament length are needed in order to relate the measure

data to a polymer network model. The microrheology data for each type of actin solution, but without adding polymerization buffer, are also a necessary control. Two point microrheology would seem essential to control for issues due to probe/solution effects. These issues of filament length would also affect the aster formation studies.

Figure 4 and the discussion of counterions is very interesting and provides a new model to understand the role of different actin isoforms. It would be good to include the other isoforms studies here in the plots of Fig 4.

The data showing similar effects of crosslinkers are also clear and valuable.

Smaller points:

1. Section: Filament Length Distribution.

The statement “The filament lengths of a semiflexible chain forming an entangled network are expected to have only a minor effect on the plateau modulus” is relevant only for networks where the mesh size is much smaller than the length of the filament, and that is not the condition for microrheology.

The text states that “all tested isoforms display median filament lengths of around 10 μm ” but table 1 and the SI show that most filaments are significantly shorter.

Fig 2A. The network of beta actin looks sparser in this image. Can network heterogeneity be analyzed from the images?

3. What concentration of yeast actin could be attained?

The commercial nm actin is at 10 mg/ml. This suggests that conventional rheology at a range of concentrations should be feasible.

4. G' at $\omega=0.1\text{s}^{-1}$ needs more justification. How does it compare with conventional methods to obtain plateau modulus?

5. It is perhaps an overstatement that “the different isoactins are unable to replace each other without loss of function, an intracellular specialization for different mechanical tasks is inferred.” As discussed in the introduction, the work of the Kashina lab shows that the isoform differences lie in the RNA level and are due to the dynamics of translation, but not at the protein level. Are these results in dispute?

6. It is not clear what is new in figure one. The differential localization of the two actin isoforms has previously been established, as has the disposition of the acidic amino terminus of actin at the surface of the filament model.

Reviewer #2 (Remarks to the Author):

The authors perform a series of experiments on differential network formation propensities of isoactins in function of magnesium ion concentration. They show that despite similar properties of single filaments the architecture of formed network is actin sequence dependent. They focus on cytoplasmic actins that co-occur in cells albeit at different subcellular locations (as the authors elegantly confirm). This is perhaps not finally solving the long standing enigma on the extreme conservation of isoactins and their subtle variations but at least a very valuable contribution and a leap forward in understanding differential properties of the cytoplasmic actin. Recently, this discussion on differential properties has become too much an all or nothing discussion based on genetic ablation and gene editing experiments. The manuscript is a welcome biophysical study with solid experimentation on purified components contributing to essential knowledge on isoform-specific mechanical properties of F-actin filaments in higher order structures.

Suggestions for improvements

Major

Mention in the introduction the physiological concentration of magnesium ions in cells to frame why the used magnesium concentrations are relevant in a biological context.

I agree with the authors that it is interesting to evaluate the role of Mg^{2+} ions for the mechanical properties of the isoactins and the interpretation of this in terms of different types of cross-linkers (bottom page 15). These are the experiments in Figure 5 where the authors evidence cross-linking for γ -actin mediated by 15 mM Mg^{2+} similar to HMM networks (isotropic) but not for β -actin and α -actin (which follows a pattern for bundling proteins). Thus at this physiological concentrations of Mg^{2+} different networks of F-actin occur depending on the isoactin. This, however, predicts also that the interaction of these crosslinking proteins with the different network themselves might be modulate differentially by magnesium ions. I missed an experiment were this variable is studied. The cross-linking proteins are now added to F-buffer (2 mM Mg^{2+}). What happens if cross-linking proteins are added (at their current concentrations and ratios to actin used) at intermediate concentrations of magnesium ions (7.5 mM) or at 15 mM? This also in view of the data presented in S5 and S6 where Mg^{2+} -dependence on differential behavior is clearly demonstrated. Do these bundling, cross-linking proteins strengthen or weaken the observed phenomena? I think these are relevant experiments, they reflect better the cellular situation because these proteins and Mg^{2+} ions are present together in cells. This is in line with their notion of bias to a certain network architecture as mentioned in the discussion (page 19). If this bias exists, it should have an effect on binding of these actin binding proteins. With all the purified components at hand I found it strange this link to cell biology was not investigated.

Along the same lines how do the isoactins behave in presence of Ca^{2+} ? This merits attention as Ca^{2+} concentration transitions are known to affect the actin cytoskeleton albeit this is usually at μM concentrations and mediated by actin binding proteins. Yet, the affinity of calcium binding proteins is often around μM olar and involves usually 3 or 4 acidic amino acid side chains. A further difference is that acidic side chains in amino acids usually bind magnesium in a monodentate manner, whereas calcium can bind in a monodentate and a bidentate manner, likely affecting affinities. Obviously most of what we know on this is derived from intramolecular metal binding and not from the bridging interactions as proposed here. Perhaps such experiments with calcium ions are better suited for of a follow-up paper. Nevertheless, in the discussion on poly-electrolytes the other important metal ion in actin biochemistry should not be ignored and briefly discussed in terms of potential interactions with side chains of acid amino acids (see also minor comment on this).

Minor

I do not follow the authors in their reasoning in the discussion (page 18): “On the one hand the last amino acid at the N-terminus of α -actin is an aspartate and thereby more like β -actin (Fig.1D). This might already explain the bundling propensity since one would expect the most exposed residue also to be the most relevant for Mg^{2+} -complexation.” 1) It is the first amino acid that is Asp (not the last one – please correct). 2) There are no structural data that this amino acid is more exposed than the second. The fact these residues are never observed in X-ray or cryo-EM structures suggest extreme flexibility and could give rise to equal water exposure or likelihood of metal ion interaction. 3) Differential chelating properties of pairs or triplets of Asp and Glu are not considered at this point

(they are considered further up in the discussion). Is it not possible that two neighboring actin molecules each have the acidic side chains acting as a monodentate ligand? Is it possible that the stronger negative charge of Asp (being sextuple if two molecules in neighboring filaments are considered) is at play? Perhaps these two parts of the discussion can be grouped. I lack in the discussion a scheme or model explaining the difference.

Legend of figure 4: line 6 there is a + missing in Mg²⁺

Define the abbreviation MSD on first occurrence

Reviewer #3 (Remarks to the Author):

In the present manuscript the authors present a comprehensive in depth microrheology study of different actin isoforms, with emphasis on beta and gamma actin – which separate in MDCK cells into distinct structures. The authors find a significant difference in linear rheology, despite similar single filament properties, and also significant differences in the contractility behavior of acto-myosin networks and in the bundling propensity of higher concentrations of Mg⁺⁺ - the former be a direct consequence of the linear modulus the latter more subtle indication of inter-filament interactions.

The manuscript addresses a surprisingly unsettled mystery in the actin field and suggests a structure/mechanics aspect to their respective roles of the isoforms. Yet, I do have a problem in bringing the first figure together with the findings of the rest of the paper – and the quite extensive discussion of possible implications and explanations. I am a bit surprised that the authors do not look at mixtures of the isoforms and how the segregation of the filaments occurs to begin with. The authors argue in the discussion that copolymerization of the isoforms would lead to small fixed locations – I wonder why the authors do not include mixed networks of the isoforms in their analysis?

While the effects are interesting and important first step, I am not really following the arguments of their importance in cellular context. While the authors take great care to carefully argue the putative nature of their findings for the physiological function, most of it remains speculative and does not

address the observed segregation of the filaments. It may turn out that either genetic regulation or modifications of affinities of other ABP (like profilin, cofilin or Arp2/3 to name some major once not considered) would indeed explain already all effects found in cells – without any mechanical role but rather biochemical activation aspects playing the role. The authors elude to this point in the last paragraph, yet it remains a bit confusing to sort out all raised points about the mechanics. I assume the authors want to point out that the mechanics is only a signature of the molecular difference, not the functional role.

The minor points are:

I would strongly prefer reconstituted actin networks instead of Artificial actin networks

Active network mechanics is confusing, as active and passive microrheology was original describing different methods – here the effect of active motors on the structure of networks is meant. And indeed the authors do not show the activity per se put the result of the activity of than again static structure.

In place of ref 63 the original publication should be cited (PNAS, 104, 51, 20199-) where nonlinear response curves are used to argue for the presence of weak interactions in entangled solutions. Actually, I would assume that such nonlinear measurements would have been better suited to quantify the molecular difference of the filaments.

The concept of contractility-connectivity dependence was actually first presented by Köhler et al. Nature Materials 2011 and PLoS ONE 6(8): e23798 and than further extended to a phase diagram by McKintosh and Koenderink.

Reviewer #1 (Remarks to the Author):

This study addresses the important and still unresolved issue of why cells express different actin isoforms that differ only in a few acidic residues at their N- termini. The study employs a range of state-of-the-art methods to use microbiological methods to obtain shear moduli of networks formed by the different isomers, as well as imaging to determine their filament lengths, persistence lengths, and reorganization due to cross linkers, magnesium ions, and myosin motors. A difference is noted between gamma and beta actins, with the conclusion that these differences in network forming properties help explain why genes coding for these different actin isoforms are not interchangeable themselves. This protein-based interpretation is a contrast to the recent model suggesting that it is the RNA coding for these different isoforms, which are localized differently and translated at different rates, and not the protein structure itself that accounts for the retention of these actin isoforms during evolution.

Several technical issues, largely regarding the microbiology studies, need to be addressed before the significance of this study can be determined.

*There is also an issue of novelty in the comparison of these different isoforms. The paper cites a review article by Ira Hermans but does not cite this lab's very closely related study, (Allen PG ... Herman IM. Phalloidin binding and rheological differences among actin isoforms. *Biochemistry*. 1996;35(45):14062-9. doi: 10.1021/bi961326g) which also measured the shear moduli of networks formed by beta and gamma actin, as well as a range of different alpha actins, along with the lengths and persistence lengths of the filaments used to form the networks. It is not clear why this work is not cited, as it covers much of the same ground and also finds that gamma actin is more efficient than beta actin at making elastic networks.*

We appreciate the referee's remark regarding the study carried out by Allen and colleagues (*Biochemistry* 1996, **35(45)**, 14062-9) . It indeed holds relevance in the discourse concerning the rheological properties of actin isoforms. Our oversight in not acknowledging this significant publication is regrettable, and we have incorporated this work now into the manuscript. We did, however, cite previous work from the authors but missed to mention this one. Thank you for bringing this to our attention. Regarding the originality of our work, it's important to highlight that we have identified several decisive distinctions between their study and ours, including:

Of utmost significance, the actin employed by Allen and colleagues comprises blends of isoforms, as opposed to singular isoforms. This prevents making straightforward claims regarding the diverse viscoelastic characteristics exhibited by different isoforms.

The compositions of the utilized actin mixtures, including any estimated isoform ratios where applicable, along with their source and the referenced purification method, are outlined as follows:

- Skeletal muscle α -actin (rabbit): Allen and colleagues cite Chaponnier and colleagues (*JCB* 1995, **130(4)**, 887-95) referring to Spudich and Watt (*J. Biol. Chem.* 1971, **246(15)**, 4866-71) regarding the actin extraction. The commercial α -actin from Cytoskeleton, Inc. (Denver, CO) employs the protocol from Pardee and Spudich (*Methods Enzymol.* 1982, **85**, 164-81), which is heavily based on Spudich and Watt (1971) and can thus be considered to produce pure skeletal muscle α -actin.
- Skeletal muscle α -actin (chicken myofibrils, presumably from gizzards): The cited method originates from Shuster and Herman (*JCB* 1995, **128(5)**, 837-48), which used modified protocols from Herman and Pollard (*JCB* 1979, **80(3)**, 509-20) and Spudich and Watt (1971). From Shuster and Herman it can be inferred that myofibrils from chicken gizzards were used for the creation of acetone powders. However, reports from North and colleagues (*J. Cell. Sci.* 1994, **107**, 445-55) and Vandekerckhove and Weber (*J. Mol. Biol.* 1978, **126(4)**, 783-802) state that the primary actin present in chicken gizzard, i.e. around 80 %, is smooth muscle γ -actin. Hence, it remains uncertain whether the reported results by Allen and colleagues genuinely reflect pure chicken skeletal muscle α -actin or instead a blend of multiple isoforms.
- Smooth muscle α -actin (bovine aorta): They used the actin extraction method described in Strzelecka-Golaszewska and colleagues (*BBA* 1985, **828(1)**, 13-21), which details that the extracted actin is a composition of 77 % smooth muscle α -actin and 23 % smooth muscle γ -actin.

- Smooth muscle γ -actin (chicken gizzard): Allen *et al.* cite the method published in Strzelecka-Golaszewska and colleagues (*Eur. J. Biochem.* 1980, **104**, 41-52). However, as mentioned before, this actin most likely represents a mixture of several different isoactins (80%, i.e. smooth muscle γ -actin and 20% cytoplasmic β -actin, as shown by North *et al.* (1994) and Vandekerckhove and Weber (1978)).
- Cytoplasmic β -actin (bovine erythrocyte): The cited method is from Shuster and Herman (1995), which applies the extraction protocol from Puszkun and colleagues (*Biochim. Biophys. Acta Biomembr.* 1978, **513(2)**, 205-220) on acetone powder of bovine erythrocytes. It is not clear, however, whether the extracted actin is a mixture of isoforms or not. Pinder and Gratzer (*JCB* 1983, **96(3)**, 768-775) show that the actin content of human erythrocytes solely consists of cytoplasmic β -actin, but whether the same holds true for bovine erythrocytes remains speculative.

Nevertheless, even in light of these considerable distinctions, the authors observed, akin to our findings, that skeletal muscle α -actin's and smooth muscle γ -actin's behavior leans more towards that of an elastic substance rather than cytoplasmic β -actin. However, a definitive direct comparison with cytoplasmic γ -actin was not furnished.

In summary, we demonstrate that subtle distinctions in amino acid sequence and composition play a crucial role in determining network characteristics such as mechanics and bundling propensity. Consequently, we firmly believe that our research offers novel data and valuable insights.

The major issue in understanding the significance of the microbiological data is the relationship between the mesh size and the average filament length. At the high dilution needed for microbiology, the mesh size is stated to be on the order of a micron, or the size of the probe (2 μm), but the average filament length for all isoforms, and especially for the non muscle actin, is very close to the same distance or even smaller. It is hard to see how a filament system with average length 0.85 microns can make an elastic network with the mesh size of 1 micron. Even if the mesh size is small enough for filaments of this length to overlap each other, the model of a network of semi flexible polymers, and especially as stated here entangled polymers, cannot apply. At a minimum, conventional measurements of elastic modulus at protein concentrations large enough so that the mesh size is much larger than the average filament length are needed in order to relate the measure data to a polymer network model. The microrheology data for each type of actin solution, but without adding polymerization buffer, are also a necessary control. Two point microrheology would seem essential to control for issues due to probe/solution effects. These issues of filament length would also affect the aster formation studies.

We agree with the referee that our presentation of the filament length distribution was misleading and in need of clarification. In a nutshell, albeit the number of short filaments is large most actin is incorporated in the larger filaments. This is best reflected in a more reasonable averaging of the histograms as outlined below. Such a method was introduced by Burlacu and colleagues (*Am. J. Physiol. Cell Physiol.* 1992, **262(3)**, 569-577) with the weighted mean length L_w :

$$L_w = \frac{\sum_i N_i X_i^2}{\sum_i N_i X_i}.$$

Here the filament lengths are divided into intervals $I_i = i \times 0.5 + [-0.25, 0.25[$ ($i \in \mathbb{N}$) with mean $X_i = i \times 0.5 - 0.25$, both in dimension μm . The N_i represent the counts per interval I_i . This weighting of the length distribution properly reflects that longer filaments contribute with a bigger weight to network properties as more actin is bound within them. From these results (see table below) we can safely assume that the overwhelming majority of actin consists of monomers bound in long filaments (larger than $5\mu\text{m}$). We show the respective histograms in the supporting information and clarified the issue in the text.

These values (table 1) are also in good agreement with the findings of Burlacu *et al.* (1992).

Consequently, our data suggests that the filament length distributions align with the expected characteristics for F-actin solutions prepared using our protocol. Additionally, we shortened F-actin filaments with the aid of gelsolin to reaffirm that our measurements are indeed focused on network

Table 1: Number average length L_n and weight average length L_w with their respective standard deviations, as defined in Burlacu *et al.* (1992).

	α	β	γ	mix
L_w (μm)	5.2 ± 3.0	10.3 ± 6.7	12.8 ± 10.9	3.1 ± 2.5

properties. In Fig. 1, the box plots illustrate how the plateau modulus (G' at 0.1 Hz) and fluidity (G''/G' at the G_0 position) of α -actin vary with different filament lengths: 1 μm , 5 μm , and 17.5 μm . As anticipated based on both theoretical predictions (Maggs, *Phys. Rev. E* 1997, **55**, 7396) and prior experimental findings in the context of 1P microrheology experiments (Weitz and colleagues, *Phys. Rev. Lett.* 2006, **96**, 118104), our data reveals highly comparable results for G_0 when comparing untreated F-actin networks to gelsolin-shortened ones with filament lengths of 17.5 μm and 5 μm . As filament length decreases down to 1 μm , the system's fluidity begins to rise, ultimately reaching its peak for the unpolymerized solution (G-actin), as anticipated.

Figure 1: Box plots for α -actin at a concentration of 24 μM . Shown are microrheology measurements done with hVPT on networks with different filament lengths (an unmodified distribution, three gelsolin-controlled distributions with filament length averages of 17.5 μm , 5 μm and 1 μm as well as an unpolymerized G-actin solution). The first plot shows plateau moduli, the second loss moduli and the the third the fluidity (G''/G' at the position of G_0) to compare elastic and viscous behavior of the actin networks.

Moreover, we performed 2P-microrheology as requested and confirmed the expected scaling of G_0 versus actin concentration (see below) to verify that our networks are composed of entangled F-actin with lengths exceeding the bead size. According to Schmidt's estimation (*Macromolecules* 1989, **22**, 3638):

$$\Rightarrow \xi = \sqrt{\frac{3}{N_A \cdot c \cdot l_{\text{actin}}}} \approx 390 \text{ nm},$$

the mesh diameter is roughly five times smaller than the probe size (2 μm). This fulfills the condition for the probe to remain trapped within the F-actin mesh, while still placing it within the relevant length scales dictated by the network.

Another distinctive characteristic of F-actin networks is the relationship between the plateau modulus and the actin concentration, as established in previous studies (Isambert and Maggs (*Macromolecules* 1996, **29**(3), 1036-40); Maggs (1997); Hinner and colleagues (*Phys. Rev. Lett.* 1998, **81**(12), 2614)). Based on prior research, we anticipate a scaling behavior of $G_0 \sim c^{7/5}$ for semi-dilute, semi-flexible polymers (Isambert and Maggs (1996); Maggs (1997); Hinner *et al.* (1998)).

To observe this scaling behavior, the network properties should predominantly involve filaments with a contour length approximately equal to or greater than the actin's persistence length (which is approximately 10 μm). In contrast, if the filaments are shorter than the mesh size or not significantly entangled, we would anticipate a scaling behavior resembling that of rigid rods ($G_0 \sim c$, Doi and Edwards "The Theory of Polymer Dynamics" (1988)). Additionally, for shorter, non-entangled filaments, we might expect predominantly viscous properties without a distinct frequency regime where $G' > G''$.

In Fig. 2, the graphs illustrate the concentration dependence of the plateau modulus for the different isoforms. This behavior aligns with the characteristic scaling observed in entangled networks of

semi-flexible polymers, reaffirming our measurement of bulk properties in the samples. We have incorporated this data into the manuscript.

Figure 2: Initial G-actin concentration dependency of the plateau moduli for α -, β - and γ -actin networks. α -actin is shown in blue, β -actin in orange, γ -actin in yellow and the straight black line represents a slope of $c_{\text{actin}}^{7/5}$, which is the expectation for the concentration dependency of G_0 in entangled networks of semi-flexible filaments, derived from the tube model.

We also conducted a comparison between one-particle microrheology (1P-MR) and two-particle microrheology (2P-MR) as recommended by the referee (refer to Fig. 3). This comparison was applied to both α -actin and γ -actin. Our findings largely align with the results previously reported by Gardel *et al.* (*Phys. Rev. Lett.* 2004, **91**, 158302), where they observed that, the plateau moduli determined from the MSD at $\tau > 10$ s using 1P and 2P-MR are almost indistinguishable. Significantly, as the particle size increases (diameter: d), the 1P and 2P MSDs approach each other more closely as the length scales converge, i.e. at $L = d$ the two different MSD match. Our data clearly indicates that $L > d$ as the MSD deviate substantially at $\tau < 10$ s.

Concerning the behavior of the two-particle mean squared displacement (2P-MSD) in relation to its frequency dependence and magnitude, it displays a distinct qualitative pattern when contrasted with the one-particle MSD (1P MSD) (see Fig. 3). For instance, at a time lag of $\tau = 0.1$, s, the 2P MSD is almost an order of magnitude smaller than the 1P MSD due to additional viscoelastic relaxation. Moreover, the scaling behavior of 2P MSD is different, with $\text{MSD} \propto \tau^{1/2}$, in contrast to the less pronounced variation observed in the 1P-MSD with changing τ . At lower frequencies, the 1P-MR shows an elastic plateau in samples where $L > d$, attributed to steric hindrance caused by filament entanglement at the entanglement length. In our manuscript, we demonstrate that this plateau scales with actin concentration as predicted for entangled F-actin networks (see Fig. 2). Eventually, this same plateau is reached by 2P-MR, and we deliberately selected a frequency of approximately 10 Hz for this purpose to determine the value of G_0 .

It's worth noting that, consistent with observations by Liu and colleagues (*Phys. Rev. Lett.* 2006, **96**, 118104), both 1P-MR and 2P-MR-derived MSDs converge at lower frequencies. However, it's important to acknowledge that the data obtained from 2P-MR is statistically less reliable than that from 1P-MR, particularly at the longer time scales relevant to our study. Consequently, we primarily relied on 1P-MR in our study.

Figure 3: Comparison of one-particle (red) and two-particle (blue) MSDs in 24 $\mu\text{M}/\text{ml}$ F-actin solution with particle radius $a = 2 \mu\text{m}$. The individual MSDs are depicted in a transparent color, while the average is presented by solid symbols of the same color. Convergence is observed for lag times larger than 10 s. **A** α -actin. **B** γ -actin.

Therefore, we present the following arguments:

1. At short times, there is a distinction between the 1P-MSD and 2P-MSD, indicating that $L > d$, which implies the formation of entangled networks.
2. Beyond $\tau > 10, \text{s}$, the 1P- and 2P-MSDs converge, allowing us to determine the plateau modulus G_0 .
3. Gardel *et al.* showed that at low frequencies, the plateau moduli obtained from 1P and 2P-MR are nearly indistinguishable.
4. Given that 2P-MR exhibits considerably more noise than 1P-MR, our comparison of actin isoforms at extended time scales and lower frequencies was conducted exclusively using 1P-MR.

Furthermore, we conducted a comparative analysis of our holographic video particle tracking (hVPT) data with results obtained through various other methods. We have carried out the analysis solely for α -actin due to the substantial sample volume requirements for most of these techniques. These methods include active optical tweezers measurements (OT, microrheology), dynamic light scattering (DLS, microrheology), diffusing wave spectroscopy (DWS, macrorheology), and Rheometer measurements (macrorheology), all performed under highly similar conditions to our hVPT setup. The comparison is presented in Fig. 4 for different α -actin concentrations and Tab. 2 (a segment of the figure centered around a concentration of $20 \mu\text{M}$). The figure illustrates a strong alignment among all microrheological methods and a somewhat smaller yet still notable alignment between micro- and macrorheological methods, as anticipated. These findings closely mirror prior observations and lend further credibility to our hVPT data.

Table 2: Comparison of different rheological methods around a concentration of $20 \mu\text{M}$ α -actin. Shown are mean shear moduli \bar{G}_0 from Fig. 4.

c_{actin} (μM)	Method	\bar{G}_0 (Pa)
20	hVPT	0.15 ± 0.01
17, 19 and 20	OT	0.16 ± 0.07
20	DLS	0.10 ± 0.01
15	DWS	0.44 ± 0.04
19	Rheometer	0.55 ± 0.23

Figure 4: An evaluation of various micro- and macrorheological techniques for assessing the mechanical characteristics of reconstituted F-actin networks is presented. The graph displays the average plateau moduli for α -actin networks across concentrations ranging from 5 to 40 μM . The primary method employed, holographic video particle tracking (1 particle), is represented by blue squares, while active optical tweezers (1 particle) are denoted by orange diamonds. Dynamic light scattering results are indicated by cyan squares, diffusing wave spectroscopy by black triangles, and measurements conducted with a rotating disc parallel plate rheometer are depicted as red stars.

Figure 4 and the discussion of counterions is very interesting and provides a new model to understand the role of different actin isoforms. It would be good to include the other isoforms studies here in the plots of Fig 4.

We concur with the referee and have incorporated their suggestion to expand Figure 4 including the data for α -actin and NM-actin as requested.

Smaller points:

Filament Length Distribution.

The statement “The filament lengths of a semiflexible chain forming an entangled network are expected to have only a minor effect on the plateau modulus” is relevant only for networks where the mesh size is much smaller than the length of the filament, and that is not the condition for microrheology. The text states that “all tested isoforms display median filament lengths of around 10 μm ” but table 1 and the SI show that most filaments are significantly shorter.

As discussed above, we have taken the necessary steps to address and rectify the identified text passage.

Fig 2A. The network of beta actin looks sparser in this image. Can network heterogeneity be analyzed from the images?

Overall, this is not the general scenario, and in this specific instance, the apparent disparity in network structures between the displayed CLSM image for β -actin and γ -actin is more attributable to the selection of an inadequate example rather than a quantifiable difference between the isoforms. To address this concern, we conducted an analysis of CLSM images depicting networks formed by various actin isoforms, wherein we measured intensity decorrelation along horizontal lines in pixels as a metric for discerning distinctions. In light of the consistent data obtained, we opted to replace the image representing β -actin with a more representative one to prevent any potential misinterpretation.

As depicted in Figure 5, the standard deviations of texture correlation for each isoform exhibit sub-

Figure 5: Correlation of pixel intensities in CLSM images for each isoform. The shaded areas around the plotted lines represent the standard deviation.

stantial overlap. Nevertheless, it is worth noting that there are additional methods available to assess heterogeneity, but carrying out such experiments would require sparser networks at lower actin concentrations.

3. *What concentration of yeast actin could be attained? The commercial nm actin is at 10 mg/ml. This suggests that conventional rheology at a range of concentrations should be feasible.*

Although it would be desirable to incorporate traditional rheology experiments as a means of comparison to ensure that we are probing bulk properties accurately, the required sample volume for an equivalent number of measurements surpasses our current supply of both γ - and β -actin. Nonetheless, in response to the aforementioned question, we carried out a range of rheological experiments involving α actin. These experiments essentially demonstrate that bulk rheology consistently yields higher shear moduli in comparison to microrheology.

As per your request, we have additionally utilized two- particle microrheology to demonstrate that the overall trend remains consistent also in bulk rheology assessments. The averaged mean square displacement is found to be only half as large as observed in single-particle microrheology, aligning with the predictions made by Weitz and colleagues (Phys. Rev. Lett. 2006, **96**, 118104).

Given that two particle rheology and one point microrheology essentially offer equivalent information about actin networks considered here and probed with a particle size of 1 μm , we will continue to focus on one point microrheology throughout the remainder of the manuscript due to its significantly superior signal-to-noise ratio.

4. *G' at $\omega=0.1\text{s}^{-1}$ needs more justification. How does it compare with conventional methods to obtain plateau modulus?*

Table 3 shows G' for the three isoforms over a wide frequency range as well as for the minima of G'' and the loss tangent $\tan \delta = \frac{G''}{G'}$, which can be utilized to determine G_0 (Liu *et al.* (Polymer 2006, **47(13)**, 4461-79)). We opted for a frequency of 0.1 Hz because the MSDs exhibit a distinct plateau at the corresponding time lag, and the two-particle MSDs align with the behavior of the one-particle MSD (see Fig. 3). Furthermore, our results for both β -actin and γ -actin display no noteworthy deviation compared to the outcomes obtained using the other two conventional methods, which also exhibit minima within the range of approximately 0.05 to 0.1 Hz within this concentration range. α -actin shows small deviations within the error range and much smaller than the difference between isoforms. The use of this fixed frequency allows for better signal-to-noise ratios also because the more error prone G'' data neither have to be smoothed nor fitted to a polynomial. Hence, the manipulation of data by smoothing is minimized. Figure 2 in the manuscript displays the average mean square displacement of the pure isoforms, providing additional illustration that the variations

in mechanical properties persist across the entire frequency spectrum.

Table 3: Storage moduli for the analyzed isoforms at different frequencies to determine G_0 .

Frequency (Hz)	α - G' (mPa)	β - G' (mPa)	γ - G' (mPa)
1	115 ± 20	9 ± 1	40 ± 3
0.5	100 ± 16	8 ± 1	38 ± 3
0.1	79 ± 11	7 ± 1	39 ± 3
0.05	76 ± 10	7 ± 1	39 ± 3
0.01	71 ± 10	7 ± 1	40 ± 3
min G''	75 ± 10	7 ± 1	39 ± 3
min $\tan \delta$	75 ± 10	7 ± 1	39 ± 3

5. *It is perhaps an overstatement that “the different isoactins are unable to replace each other without loss of function, an intracellular specialization for different mechanical tasks is inferred.” As discussed in the introduction, the work of the Kashina lab shows that the isoform differences lie in the RNA level and are due to the dynamics of translation, but not at the protein level. Are these results in dispute?*

We acknowledge the reviewer’s observation that the highlighted statement was an overstatement, and we have since revised it accordingly. It is essential to clarify that we do not consider our findings to be in conflict with the research conducted by the Kashina lab. Instead, we view our work as contributing to a different layer of understanding the complex interactions between actins and other proteins. In this regard, there remain certain perplexing aspects related to isoactins that the recent studies from the Kashina lab do not fully address. For example, disparities in isoactin behavior concerning polymerization dynamics (Bergeron *et al.* (*J. Biol. Chem.* 2010, **285**(21), 16087-95)) or interactions with myosin (Müller *et al.* (*PLoS One* 2013, **8**(7), e70636)) have been observed in reconstituted systems. Consequently, we see the Kashina lab’s findings as complementary to our results. Our discovery indicates that β -actin has a strong tendency to form bundles and predominantly locates itself in stress fibers, while γ -actin tends to create a mesh-like structure and predominantly resides in the cortex. This observation suggests a close correlation between mesoscopic network architecture and amino acid sequence.

6. *It is not clear what is new in figure one. The differential localization of the two actin isoforms has previously been established, as has the disposition of the acidic amino terminus of actin at the surface of the filament model.*

We agree with the reviewer regarding the lack of novelty in Fig. 1 of the manuscript. We replaced panels with redundant information and added a small scheme to visualize the isoform-specific viscoelastic network properties.

Figure 6: β -actin (green) and γ -actin (red) are locally segregated in MDCK II cells. β -actin is exclusively associated with basal stress-fibers (A, left panel), while γ -actin is predominately localized at the apical cortex (A, right panel). An orthogonal view (A, bottom panel) shows that the cell-cell-boundaries mainly consist of β -actin [Dug+09] (scale bars: 20 μ m). (B, C) The main differences (green) between the isoforms are concentrated at the N-terminus, which is located at very exposed positions along the filament (adapted from [Con08]). (C) Structures of actin isoforms measured via cryo electron microscopy. For the shown protein structures the Protein Database (PDB) entries 3B5U, 8DNH and 8DNF were used. (D, E) The schematics depict entangled isoactin filament networks (the green structures are β -actin filaments, red structures are γ -actin) with a microparticle (light blue) and different trajectories (black) indicating the measurement of the isoform-specific viscoelastic network properties. (F) Fluorescence micrograph of an γ -actin network (12 μ M at 30 mM Mg^{2+} stained with Alexa-Fluor-488-phalloidin) together with embedded microparticles (scale bar: 10 μ m).

Reviewer #2 (Remarks to the Author):

The authors perform a series of experiments on differential network formation propensities of isoactins in function of magnesium ion concentration. They show that despite similar properties of single filaments the architecture of formed network is actin sequence dependent. They focus on cytoplasmic actins that co-occur in cells albeit at different subcellular locations (as the authors elegantly confirm). This is perhaps not finally solving the long standing enigma on the extreme conservation of isoactins and their subtle variations but at least a very valuable contribution and a leap forward in understanding differential properties of the cytoplasmic actin. Recently, this discussion on differential properties has become too much an all or nothing discussion based on genetic ablation and gene editing experiments. The manuscript is a welcome biophysical study with solid experimentation on purified components contributing to essential knowledge on isoform-specific mechanical properties of F-actin filaments in higher order structures.

Major

Mention in the introduction the physiological concentration of magnesium ions in cells to frame why the used magnesium concentrations are relevant in a biological context.

Certainly, it is crucial to include the physiological magnesium concentration to enable the reader to assess the significance of our experiments conducted with specific magnesium ion concentrations in a biological context. As indicated by Moomaw and Maguire (*Physiology* 2008 **23**, 275-85) and others (see below in the follow-up question), the concentration range of magnesium ions in the cytosol typically falls within the range of 15-25 mM (10-20 mM in other sources, see below). However, within the cytosol, most of the Mg^{2+} ions are tightly bound to ATP, along with other phosphonucleotides, as well as to numerous enzymes. Mg^{2+} plays a vital role in all cells, serving as a fundamental structural component in ribosomes and membranes, and acting as an indispensable co-factor for ATP within the catalytic sites of various enzymes. We have incorporated this information into the introduction section for clarity.

I agree with the authors that it is interesting to evaluate the role of Mg^{2+} ions for the mechanical properties of the isoactins and the interpretation of this in terms of different types of cross-linkers (bottom page 15). These are the experiments in Figure 5 where the authors evidence cross-linking for γ -actin mediated by 15 mM Mg^{2+} similar to HMM networks (isotropic) but not for β -actin and α -actin (which follows a pattern for bundling proteins). Thus at this physiological concentrations of Mg^{2+} different networks of F-actin occur depending on the isoactin. This, however, predicts also that the interaction of these crosslinking proteins with the different network themselves might be modulate differentially by magnesium ions. I missed an experiment were this variable is studied. The cross-linking proteins are now added to F-buffer (2 mM Mg^{2+}). What happens if cross-linking proteins are added (at their current concentrations and ratios to actin used) at intermediate concentrations of magnesium ions (7.5 mM) or at 15 mM? This also in view of the data presented in S5 and S6 where Mg^{2+} -dependence on differential behavior is clearly demonstrated. Do these bundling, cross-linking proteins strengthen or weaken the observed phenomena? I think these are relevant experiments, they reflect better the cellular situation because these proteins and Mg^{2+} ions are present together in cells. This is in line with their notion of bias to a certain network architecture as mentioned in the discussion (page 19). If this bias exists, it should have an effect on binding of these actin binding proteins. With all the purified components at hand I found it strange this link to cell biology was not investigated.

We fully agree, that the presented experiments must approach physiological conditions as far as possible since they are supposed to investigate integral aspects of cellular mechanics. The same way we argue for a careful choice of actin isoforms in reconstituted model systems for rheology, the arguably most important divalent cation regarding cell mechanics should also be taken into account in a concentration regime comparable to cellular conditions. While it is true that total cellular magnesium concentrations are reported between 10 and 20 μ M (Castaneda *et al.* (*J. Phys. Chem. B*., 2018, **122**(14), 3826-35); Romani (*Front. Biosci.*, 2007, **12**, 308-31); Romani (*Arch. Biochem. Biophys.*, 2011, **512**(1), 1-23); Gotoh *et al.* (*Brain Res.*, 1999, **828**(1-2), 163-8); Griswold and Pace (*Exp. Cell Res.*, 1956, **11**(2), 362-7); Günther (*Magnesium*, 1986, **5**(2), 53-9)), as far as we were able to find, free Mg^{2+} concentration are much closer to 2 μ M (Romani and Scarpa (*Arch.*

Biochem. Biophys., 1992, **298**(1), 1-12); Apell *et al.* (*Biochemistry*, 2017, **56**(7), 1005-16); Günther (*Artery*, 1981, **9**(3), 167-81)) and thereby to the concentration we chose. Nevertheless we added new experiments in the suggested concentration regime, since the simultaneous interaction of actin filaments, actin binding proteins and divalent cations is also not well understood while obviously being of high physiological relevance. Additional experiments at elevated cation concentrations possibly enable the understanding of weak interactions by amplification.

Figure 7: Crosslinking of α -actin at elevated Mg^{2+} concentrations. No bundling is observable at chosen actin- and crosslinker concentrations (12 μ M actin, 120 nM crosslinkers and 2 mM Mg^{2+}). It makes sense that bundling occurs at elevated concentrations of actin and (bundling-) crosslinkers. The effect of a combination of crosslinkers (at concentrations below the bundling transition) and elevated Mg^{2+} concentration (below the counterion condensation bundling transition) is less well known. It can illuminate the interaction of different isoforms with divalent cations by forcing increased interaction through crosslinkers. Shown are α -actin networks crosslinked with different crosslinkers (A and D: α -actinin, B and E: fascin and C and F: HMM) at different Mg^{2+} concentrations (upper panels are at 7.5 mM Mg^{2+} , lower panels at 15 mM Mg^{2+}). Scale bars are 10 μ m.

Figure 8: Crosslinking of β -actin at elevated Mg^{2+} concentrations. No bundling is observable at chosen actin- and crosslinker concentrations ($12\ \mu\text{M}$ actin, $120\ \text{nM}$ crosslinkers and $2\ \text{mM}$ Mg^{2+}). Shown are β -actin networks crosslinked with different crosslinkers (A and D: α -actinin, B and E: fascin and C and F: HMM) at different Mg^{2+} concentrations (upper panels are at $7.5\ \text{mM}$ Mg^{2+} , lower panels at $15\ \text{mM}$ Mg^{2+}). Scale bars are $10\ \mu\text{m}$.

Figure 9: Crosslinking of γ -actin at elevated Mg^{2+} concentrations. No bundling is observable at chosen actin- and crosslinker concentrations ($12\ \mu\text{M}$ actin, $120\ \text{nM}$ crosslinkers and $2\ \text{mM}$ Mg^{2+}). Shown are γ -actin networks crosslinked with different crosslinkers (A and D: α -actinin, B and E: fascin and C and F: HMM) at different Mg^{2+} concentrations (upper panels are at $7.5\ \text{mM}$ Mg^{2+} , lower panels at $15\ \text{mM}$ Mg^{2+}). Scale bars are $10\ \mu\text{m}$.

As suggested by the referee, we now also examined crosslinked, reconstituted actin networks at elevated Mg^{2+} concentrations. The initial expectation is that bundling would occur at lower concentrations due to cooperative crosslinking, effectively resulting in a higher effective crosslinker concentration (of bundling crosslinkers such as α -actinin and fascin in contrary to HMM), which should lower the threshold Mg^{2+} concentration required for bundling. Bundling occurs when crosslinking proteins or cations stabilize filaments (which are strongly negatively charged) in close proximity, either parallel or anti-parallel, thereby counteracting thermal fluctuations and charge repulsion. Consequently, the combination of divalent cations with bundling crosslinkers should reduce the critical concentrations required for bundling.

We observed that at Mg^{2+} concentrations exceeding 7mM, either bundling or partial bundling occurred, resulting in localized non-uniformity and a departure from the continuum assumption in microrheology. This makes the determination of viscoelastic properties challenging. Moreover, the presence of partial bundling significantly reduces the significance of bundling parameters. In Figures 7, 8 and 9, typical confocal images are presented, illustrating the bundling of cross-linked networks formed by various isoforms. Interestingly, we noted that γ -actin exhibits a higher resistance to bundling when compared to β -actin under identical cross-linker and magnesium concentrations. Notably, γ -actin does not undergo structural changes in the network with increasing Mg^{2+} concentrations (from 2 to 7.5 and 15 mM). This strongly suggests a stabilization of the network in the form of interfilament interactions, transmitted by Mg^{2+} ions, like already inferred from the Mg^{2+} -bundling properties.

In contrast, for β -actin, all three crosslinkers induce a noticeable increase in bundling. This observation aligns with the idea that these filaments may have fewer or weaker interactions with Mg^{2+} . Consequently, the networks do not experience additional stabilization against protein-based bundling or counterion condensation, leading to the formation of bundles at lower concentrations when each crosslinker is introduced individually. This observation is also consistent with our prior findings and suggests a more robust, complexation-based inter-filament interaction for γ -actin.

Furthermore, a more detailed examination reveals a greater increase in bundling for α -actinin compared to fascin, which is considerably smaller in size (Winkelman et al. (*Curr. Biol.*, 2016, **26**(20), 2697-706)) and therefore needs to overcome more repulsion. On the other hand, we observed the least increase in bundling for HMM, which appears to play a stabilizing role in β -actin networks at 7.5 mM Mg^{2+} .

We express our gratitude to the referee for bringing these experiments to our attention. We have incorporated these findings into the main body of the text and included a corresponding figure in the supplementary information.

Along the same lines how do the isoactins behave in presence of Ca^{2+} ? This merits attention as Ca^{2+} concentration transitions are known to affect the actin cytoskeleton albeit this is usually at μM concentrations and mediated by actin binding proteins. Yet, the affinity of calcium binding proteins is often around μmolar and involves usually 3 or 4 acidic amino acid side chains. A further difference is that acidic side chains in amino acids usually bind magnesium in a monodentate manner, whereas calcium can bind in a monodentate and a bidentate manner, likely affecting affinities. Obviously most of what we know on this is derived from intramolecular metal binding and not from the bridging interactions as proposed here. Perhaps such experiments with calcium ions are better suited for of a follow-up paper. Nevertheless, in the discussion on poly-electrolytes the other important metal ion in actin biochemistry should not be ignored and briefly discussed in terms of potential interactions with side chains of acid amino acids (see also minor comment on this).

While our investigation primarily focuses on the role of magnesium ions in the interaction between divalent cations and cellular mechanics and structure formation, we acknowledge the significant importance of calcium ions in cellular mechanics as well. As suggested by the referee, we are currently engaged in a follow-up study to explore the impact of Ca^{2+} ions on isoform-specific bundling behavior and mechanical properties of reconstituted actin networks.

Furthermore, we concur that it is essential to mention calcium in this context. Cellular components do not operate in isolation, and the analysis and description of intricate processes like cellular mechanics should always consider as many relevant elements as realistically possible.

We generated the requested data on the Ca^{2+} dependent rheological properties and incorporated the outcome into the discussion section. Figure 10 shows the impact on calcium on the mechanical properties of the networks formed by the different isoforms - please compare also with the original figure 4(D). It gives insight into the pre-bundling interaction of F-actin with Ca^{2+} ions for different isoforms. Similarly as with Mg^{2+} -ions, stiffening of the network with increased Ca^{2+} concentration is found only for γ -actin. Figure 11 shows the bundling of all studied isoactins upon Ca^{2+} addition. At an actin concentration of 12 μM , bundles begin to appear at much lower Ca^{2+} concentrations ($\approx 10\text{mM}$, Fig. 11) compared to Mg^{2+} ($\geq 25\text{mM}$.)

Interestingly, networks formed in the presence of only 0.5mM Ca^{2+} are very soft ($\approx 10\times$ softer) compared to networks formed in the presence of 2 mM Mg^{2+} .

Hence, magnesium appears to be much more relevant for actin polymerization and network mechanics at physiological concentrations of Mg^{2+} and Ca^{2+} .

Figure 10: Effect of Calcium ions on the stiffness of reconstituted actin networks ($12 \mu\text{M}$) for different actin isoforms. Relative plateau moduli G_R (normalized with the corresponding plateau modulus for Mg^{2+} -actin networks at 2 mM Mg^{2+} , represented by 0^*) are shown against the calcium concentration ($0.5 - 5 \text{ mM}$), which is slightly above the physiological range. Lower calcium concentrations don't produce entangled (meaningfully measurable with hVPT) networks and higher concentrations lead to bundle networks (also not meaningfully measurable with hVPT). α -actin is represented by blue circles, β -actin by orange squares and γ -actin by yellow triangles.

Figure 11: Interaction of actin isoforms with Ca^{2+} ions. Shown are actin networks (α -, β - and γ -actin as indicated at $12 \mu\text{M}$ actin) at different concentrations of Ca^{2+} ions ($0, 0.5, 1$ and 5 mM from the left to the right panels; $0 \mu\text{M}$ networks are polymerized with $2 \mu\text{M Mg}^{2+}$ like usually, the others with $0 \mu\text{M Mg}^{2+}$). Scale bars are $10 \mu\text{m}$.

Once again, we extend our gratitude to the referee for recommending the execution of these experiments and for their valuable input, which we have incorporated into the manuscript.

I do not follow the authors in their reasoning in the discussion (page 18): "On the one hand the last amino acid at the N-terminus of α -actin is an aspartate and thereby more like β -actin (Fig.1D). This

might already explain the bundling propensity since one would expect the most exposed residue also to be the most relevant for Mg^{2+} -complexation.” 1) It is the first amino acid that is Asp (not the last one – please correct). 2) There are no structural data that this amino acid is more exposed than the second. The fact these residues are never observed in X-ray or cryo-EM structures suggest extreme flexibility and could give rise to equal water exposure or likelihood of metal ion interaction. 3) Differential chelating properties of pairs or triplets of Asp and Glu are not considered at this point (they are considered further up in the discussion). Is it not possible that two neighboring actin molecules each have the acidic side chains acting as a monodentate ligand? Is it possible that the stronger negative charge of Asp (being sextuple if two molecules in neighboring filaments are considered) is at play? Perhaps these two parts of the discussion can be grouped. I lack in the discussion a scheme or model explaining the difference.

We adjusted the referenced text passage. Indeed, in a very recent study conducted by Arora *et al.* (2023) the N-terminal aminoacids were resolved using data acquired by cryogenic electron microscopy. As visible from Fig. 12, the first aspartate in both α - and β -actin are spatially less exposed as the first glutamate in γ -actin. In light of this new report we have adapted our discussion accordingly and streamlined our paragraphs. As suggested by the referee, we now also provide a scheme in the discussion to illustrate our main findings.

Figure 12: Structures of actin isoforms measured via cryo electron microscopy. For skeletal muscle α -actin the Protein Database (PDB) entry 3b5u deposited by Cong *et al.* (2008) was used. For cytoplasmic β - and γ -actin the PDB entries 8DNH and 8DNF deposited by Arora *et al.* (2023) were used. The latter study also included measurements on skeletal muscle α -actin, however Arora *et al.* were not able to resolve the extreme N-terminal aminoacids.

Legend of figure 4: line 6 there is a + missing in Mg^{2+}

Amended.

Define the abbreviation MSD on first occurrence

Done.

Reviewer #3 (Remarks to the Author):

In the present manuscript the authors present a comprehensive in depth microrheology study of different actin isoforms, with emphasis on beta and gamma actin – which separate in MDCK cells into distinct structures. The authors find a significant difference in linear rheology, despite similar single filament properties, and also significant differences in the contractility behavior of acto-myosin networks and in the bundling propensity of higher concentrations of Mg^{++} - the former be a direct consequence of the linear modulus the latter more subtle indication of inter-filament interactions.

The manuscript addresses a surprisingly unsettled mystery in the actin field and suggests a structure/mechanics aspect to their respective roles of the isoforms. Yet, I do have a problem in bringing the first figure together with the findings of the rest of the paper – and the quite extensive discussion of possible implications and explanations. I am a bit surprised that the authors do not look at mixtures of the isoforms and how the segregation of the filaments occurs to begin with. The authors argue in the discussion that copolymerization of the isoforms would lead to small fixed locations – I wonder why the authors do not include mixed networks of the isoforms in their analysis?

Mixture of isoforms

Figure 13: Mechanical properties of networks formed by various mixtures of pure actin isoforms (β -actin and γ -actin) compared to NM-actin. The compositions (100/0, 85/15, 50/50, 15/85 0/100) are chosen for direct comparison with the NM-actin (85% β -actin and 15% γ -actin). Plateau moduli G_0 of the isoform mixtures and NM-actin are obtained from the purely elastic frequency regime of the viscoelastic spectra at a fixed frequency of 0.1 Hz. Solid red and blue lines represent the median and the lower and upper quartiles, respectively.

We express our appreciation to the referee for their evaluation, and we have made adjustments to Figure 1 in the original manuscript and also provide a new figure in the discussion. These modifications establish a coherent connection between this introductory figure and the findings presented in the manuscript. Indeed, a more thorough study of networks with different ratios of β - and γ -actin would be desirable, however we lack the material to do so and therefore constrained our research on networks consisting of pure isoforms. We however plan on expanding our research on such copolymerized networks for future publications.

In response to your request, we have included data from isoform mixtures (β/γ) in the Supplementary Information (compare Fig. 13). These data illustrate how the introduction of γ -actin influences the viscoelastic properties, resulting in a shift towards stiffer networks.

While the effects are interesting and important first step, I am not really following the arguments of their importance in cellular context. While the authors take great care to carefully argue the putative nature of their findings for the physiological function, most of it remains speculative and does not address the observed segregation of the filaments. It may turn out that either genetic regulation or modifications of affinities of other ABP (like profilin, cofilin or Arp2/3 to name some major once not considered) would indeed explain already all effects found in cells – without any mechanical role but rather biochemical activation aspects playing the role. The authors elude to this point in the last paragraph, yet it remains a bit confusing to sort out all raised points about the mechanics. I assume the authors want to point out that the mechanics is only a signature of the molecular difference, not the functional role.

We concur with the referee's observation that in the context of living cells, mechanical properties may be of lesser importance compared to factors like biochemical activation and gene expression patterns. Nevertheless, it is noteworthy that our findings indicate a distinct behavior: β -actin tends to associate more with stress fibers and exhibits significantly greater bundling propensity when exposed to magnesium, whereas γ -actin is predominantly located in the cortex, consistent with its mesh-like architecture and resistance against bundling. We have streamlined our discussion to enhance conciseness and present a more reader-friendly message.

The minor points are:

I would strongly prefer reconstituted actin networks instead of Artificial actin networks

The term reconstituted is obviously the accurate one here and we replaced every occurrence of the word "artificial" in this context.

Active network mechanics is confusing, as active and passive microrheology was original describing different methods – here the effect of active motors on the structure of networks is meant. And indeed the authors do not show the activity per se put the result of the activity of than again static structure.

We agree with the referee and apologize for this confusing choice of language. We edited the text accordingly and replaced the term "active mechanics" with "non-equilibrium mechanics" describing athermal, ATP-driven actomyosin motion.

In place of ref 63 the original publication should be cited (PNAS, 104, 51, 20199-) where nonlinear response curves are used to argue for the presence of weak interactions in entangled solutions. Actually, I would assume that such nonlinear measurements would have been better suited to quantify the molecular difference of the filaments.

We have included a citation to the original publication, and we are open to the possibility of conducting nonlinear measurements on the isoform in future experiments.

REVIEWERS' COMMENTS

Reviewer #1 (Remarks to the Author):

The authors have performed numerous additional experiments and added new data to the manuscript to address the previous concerns. They have adequately answered the questions addressed and significantly improved this work. I have no additional points to raise.

Reviewer #2 (Remarks to the Author):

The authors added the requested experiments investigating the mechanical properties of networks of cytosolic actin isoforms as a function of Mg^{2+} -concentration. They also addressed a discretionary revision on calcium ion transients.

Reviewer #3 (Remarks to the Author):

I thank the authors for considering the comments of all the referees thoroughly and recommend the acceptance of the manuscript.